# LLM-AutoDA:
# Large Language Model-Driven Automatic Data Augmentation for Long-tailed Problems

**Pengkun Wang**[1,2]*, **Zhe Zhao**[1,3]* **Haibin Wen**[5],
**Fanfu Wang**[6], **Binwu Wang**[1], **Qingfu Zhang**[3]†, **Yang Wang**[1,2]†
[1]University of Science and Technology of China (USTC), Hefei, China
[2]Suzhou Institute for Advanced Research, USTC, Suzhou, China
[3]City University of Hong Kong, Hong Kong, China
[5]MorongAI, Suzhou, China
[6]Lanzhou University, Lanzhou, China
{pengkun@ustc.edu.cn, zz4543@mail.ustc.edu.cn, haibin65535@gmail.com,
wangff21@lzu.edu.cn, wbw2024@ustc.edu.cn, qingfu.zhang@cityu.edu.hk,
angyan@ustc.edu.cn}

## Abstract

The long-tailed distribution is the underlying nature of real-world data, and it presents unprecedented challenges for training deep learning models. Existing long-tailed learning paradigms based on re-balancing or data augmentation have partially alleviated the long-tailed problem. However, they still have limitations, such as relying on manually designed augmentation strategies, having a limited search space, and using fixed augmentation strategies. To address these limitations, this paper proposes a novel LLM-based long-tailed data augmentation framework called LLM-AutoDA, which leverages large-scale pretrained models to automatically search for the optimal augmentation strategies suitable for long-tailed data distributions. In addition, it applies this strategy to the original imbalanced data to create an augmented dataset and fine-tune the underlying long-tailed learning model. The performance improvement on the validation set serves as a reward signal to update the generation model, enabling the generation of more effective augmentation strategies in the next iteration. We conducted extensive experiments on multiple mainstream long-tailed learning benchmarks. The results show that LLM-AutoDA outperforms state-of-the-art data augmentation methods and other re-balancing methods significantly. The code is available in https://github.com/DataLab-atom/LLM-LT-AUG.

## 1 Introduction

As a revolutionary technology, deep learning has shown a broad and significant impact on various tasks, including image classification [21], object detection [34], natural language processing [19], and many interdisciplinary research problems [40, 47]. The success of these endeavors relies heavily on the support of large-scale manually curated datasets, e.g., ImageNet [35]. However, for the convenience of training and evaluating models, most artificially constructed datasets typically follow the assumption of uniform distribution, which contradicts the real-world data distribution, i.e., long-tailed distribution. This deviation between the ideal and real distributions has resulted in many deep

---

*Pengkun Wang and Zhe Zhao contributed equally to this work.
†Corresponding authors: Qingfu Zhang and Yang Wang.

38th Conference on Neural Information Processing Systems (NeurIPS 2024).

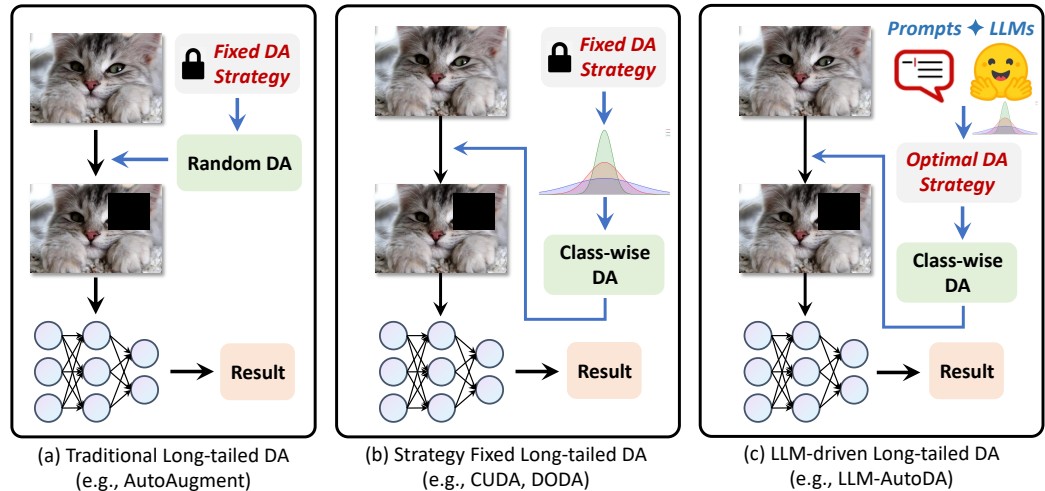

Figure 1: Different long-tailed data augmentation paradigms. (a) The traditional augmentation paradigm randomly samples augmentations from the fixed strategy. (b) The strategy fixed augmentation paradigm samples augmentations from the fixed strategy according to the data distribution. (c) The LLM-driven augmentation paradigm combines LLMs with long-tailed learning to learn the optimal augmentation strategy.

models trained on balanced datasets failing to produce satisfactory results in real-world applications, e.g., they only perform well on a few classes and ignore many vulnerable classes [4, 36].

To address the ubiquitous long-tailed problem, researchers have been continuously proposing various carefully designed research paradigms. One popular approach is to rebalance the training data by oversampling the tail classes or undersampling the head classes [6, 13]. However, this approach cannot fundamentally address the problem of insufficient global information, even though the data for the tail classes increases significantly. Another line of work focuses on designing specialized loss functions or reweighting strategies to alleviate the impact of class imbalance [11, 5, 37, 39]. However, these methods either introduce additional computational overhead or require careful manual design.

Recently, using data augmentation (DA) to improve long-tailed learning has attracted significant attention from researchers and is considered as a viable research paradigm [44]. For example, FASA [46] enhances the tail classes by generating class-level features based on Gaussian priors. Remix [8] achieves this goal through a rebalanced mixup approach. However, these DA-based methods either manipulate high-dimensional information in the feature space or directly apply traditional transformations (e.g., flipping, cropping, and rotation) to expand the training set and generate diverse data, without considering the underlying relationships between data augmentation and class classes. To avoid ineffective augmentation, some studies suggest applying different augmentations to different classes [41]. Typically, CUDA [2] improves the overall performance of models by dynamically adjusting the augmentation intensity for each class during training. Considering the issue of pseudo-boosting in augmentation, DODA [39] allows each class to choose its own suitable augmentation method, thereby avoiding weak classes being sacrificed. Unfortunately, they still have significant limitations: (i) *these strategies are often based on manually designed human knowledge and experience, which may be suboptimal for specific data and tasks.* (ii) *the search space of these strategies is often limited.* (iii) *these fixed strategies lack flexibility to adapt to changes in the data distribution during the training process.*

To address the above limitations, we leverage the recently popular large language models (LLMs) [29, 24, 45] to provide augmentation suggestions for long-tailed learning. We first designed a feasible and straightforward framework called SimpleLLM, which guides the LLM to generate augmentation strategies and apply them to long-tailed learning by providing specific prompts. Analysis revealed that the augmentation strategies generated by SimpleLLM are comparable to the effectiveness of CUDA [2] and DODA [39]. Figure 1 illustrates the differences between this framework and previous methods.

Furthermore, inspired by AutoML [17], particularly automated data augmentation [10], we propose **LLM-AutoDA**, a novel LLM-based long-tailed data augmentation framework. LLM-AutoDA leverages large-scale pretrained models to automatically search for the optimal augmentation strategies suitable for long-tailed data distributions. Specifically, we first define a broad search space that includes augmentation operations and their parameters. Then, we train an augmentation strategy generation model that generates augmentation strategies based on the class-wise statistics of the long-tailed data. This strategy is applied to the original imbalanced data to create an augmented dataset, which is used to fine-tune the underlying long-tailed learning model. Importantly, the performance improvement on the validation set serves as a reward signal to update the generation model, enabling the generation of more effective augmentation strategies in the next iteration. This process is repeated until the performance converges or the computational budget is exhausted.

Compared to previous long-tailed data augmentation methods, LLM-AutoDA offers several advantages: (i) it leverages LLMs to automatically learn augmentation strategies tailored to the characteristics of long-tailed data, without relying on human expertise. (ii) it has a more extensive search space, allowing it to discover more novel strategies. (iii) it can dynamically adjust the augmentation strategies based on performance feedback during the training process, providing flexibility and robustness. Extensive experiments on multiple mainstream long-tailed learning benchmarks demonstrate that LLM-AutoDA outperforms state-of-the-art data augmentation methods and other rebalancing techniques significantly.

The main contributions of this work are summarized as follows:

- *New augmentation paradigm*: We combine LLMs with long-tail data augmentation for the first time, providing a novel perspective for efficient long-tail learning.
- *New automated framework*: We propose a novel AutoML framework called LLM-AutoDA, which automates the search for effective data augmentation strategies for long-tailed learning, significantly reducing the cost of manually designing augmentation strategies.
- *Compelling empirical results*: We conduct extensive experiments on multiple mainstream long-tailde benchmarks, demonstrating the superiority of LLM-AutoDA compared to state-of-the-art methods.
- *In-depth analysis and insights*: We provide detailed analysis and insights into the discovered augmentation strategies, guiding future research in long-tailed learning.

## 2 Related Work

### 2.1 Long-tailed Learning (LTL)

Various approaches have been proposed to address the long-tailed learning problem, including rebalancing the training data through over-sampling the tail classes [6, 13] or under-sampling the head classes [12, 14], modifying loss functions or adjusting class weights during training [22, 11], and decoupling representation and classifier learning [18]. Among these methods, data augmentation has emerged as a promising solution for long-tailed learning [8, 9, 20, 25, 46]. The key idea is to generate additional samples for the tail classes to alleviate the data imbalance issue. Recent research has attempted to design sophisticated strategies by observing performance changes during the training process to adjust augmentation operator [39] or intensity [2]. However, these methods still rely on hand-crafted augmentation strategies that may not be optimal for the specific long-tailed data and have limited search space for discovering novel and effective strategies. In contrast, LLM-AutoDA automatically learns data augmentation strategies tailored to the long-tailed data distribution without manual design.

### 2.2 Large Language Models (LLMs)

Large language models (e.g., BERT [19], GPT [29, 30, 3], and T5 [31]) have achieved remarkable success in various natural language processing tasks. They exhibit strong generalization abilities and can be easily fine-tuned for downstream tasks with limited labeled data [16, 28]. Researchers have explored the potential of LLMs in automating algorithm design and implementation, such as generating source code [7], optimizing hyperparameters [42], and designing neural architectures [32]. However, the interaction between large and small language models and its impact on improving the

design of small models have been less explored, particularly in the context of data augmentation for long-tailed learning. LLM-AutoDA aims to bridge this gap by harnessing the knowledge and generative capabilities of large language models to discover effective data augmentation strategies tailored to long-tailed distributions automatically. By defining a rich search space of augmentation operations and training an augmentation strategy model conditioned on the class-wise statistics, LLM-AutoDA can generate adaptive and optimized augmentations specifically designed for the given long-tailed data. This novel approach opens up new possibilities for leveraging the interaction between large and small language models to improve the design of machine learning algorithms in various imbalanced learning scenarios.

# 3   LLM × LTL: Can LLMs Provide DA Strategies for Long-tailed Learning?

In this section, we attempt to analyze whether LLMs can be applied to long-tailed learning and how to implement this learning paradigm.

**SimpleLLM.** As shown in Figure 1(b), strategy fixed DA is the current mainstream paradigm for long-tailed data augmentation. It utilizes carefully designed augmentation strategies to dynamically adjust augmentation operators or intensities during the training process, allowing different classes to choose advantageous augmentation methods. In this paradigm, the key step is to design a high-quality augmentation strategy. When dealing with balanced data distributions, we often employ class-independent augmentation strategies, which apply the same data augmentation to all classes. However, as mentioned in DODA [39], when dealing with imbalanced data distributions, this class-independent augmentation strategy can potentially sacrifice certain classes, thus requiring the design of class-dependent augmentation strategies. However, the manual design process for such strategies is highly complex and costly.

Recent research has shown that LLMs can replace many manually engineered tasks [7]. Inspired by this, we first designed a simple yet efficient paradigm for generating augmentation strategies called SimpleLLM. As shown in Figure 2, we initially constructed a data augmentation-themed

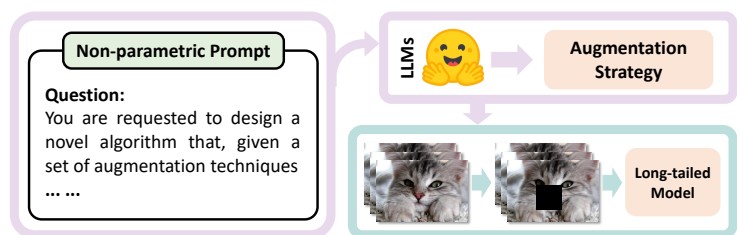

Figure 2: Strategy generation paradigm of SimpleLLM.

prompt from the perspective of prompt engineering, including task description, algorithm input, algorithm output, parameter interpretation, etc. We then input this prompt into pre-trained LLMs to generate a functional function that conforms to the prompt, i.e., an algorithm implementation containing augmentation strategies. Finally, this augmentation strategy is applied to the conventional training process of long-tailed learning. It is worth noting that this paradigm allows us to generate multiple augmentation strategies suitable for long-tailed learning at a low cost.

**Comparative Analysis.** To further validate the effectiveness of the augmentation strategies generated by this paradigm, we conducted experiments on CIFAR-100-LT (IR=100) dataset. We selected several mainstream long-tailed learning baselines and integrated SimpleLLM with them. In addition, we compared the latest state-of-the-art long-tailed DA methods, CUDA and DODA, under the same settings. The experimental results, as shown in Figure 4, reveal that SimpleLLM achieves acceptable average performance, comparable to CUDA and DODA, indicating that LLMs can be used as generators of augmentation strategies to enhance the performance of long-tailed learning.

Under this paradigm, we believe that with appropriate prompts, augmentation strategies similar to CUDA and DODA can also be generated by LLMs. In other words, within a search space, we can obtain multiple similar locally optimal strategies.

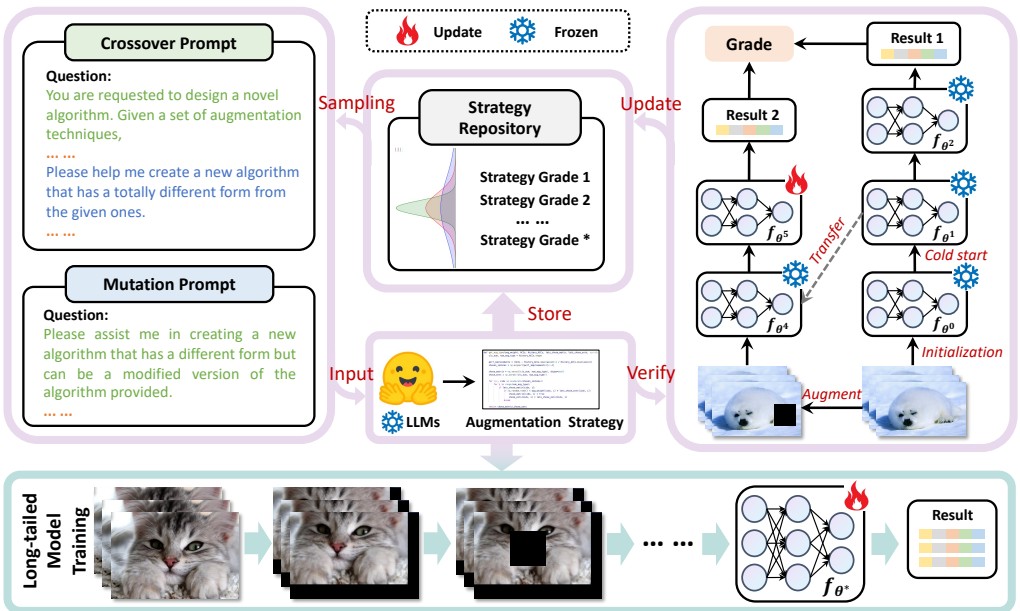

Figure 3: Overview of LLM-AutoDA. LLM-AutoDA leverages large-scale pretrained models to automatically search for the optimal augmentation strategies suitable for long-tailed data distributions.

# 4 LLM-AutoDA: A Resourceful Adviser for Long-tailed Learning

## 4.1 Framework

The overall framework of LLM-AutoDA is illustrated in Figure 3. The framework consists of two interactive modules: the LLM-based data augmentation strategy generation module and the long-tailed learning training and evaluation module.

**LLM-based Data Augmentation Strategy Generation.** As shown in Figure 3 (left, pink), to design the DA strategies automatically, LLM-AutoDA incorporates a pre-trained LLM $\mathcal{L}$ as a search operator. Using prompt engineering techniques, a series of prompt templates are designed to incorporate prior knowledge about data augmentation into the generation process of LLM. LLM generates diverse data augmentation strategies based on these prompts, including both natural language descriptions and Python code implementations. Furthermore, the generated strategies are stored in a strategy pool and interact with the long-tail learning model to search for the optimal data augmentation strategy.

**Long-tailed Learning Training and Evaluation.** As shown in Figure 3 (right, pink), LLM-AutoDA utilizes a pretrained long-tailed learning model $\mathcal{M}$ for fine-tuning on a given long-tailed distributed dataset $\mathcal{D}$. At the beginning of each training epoch, the algorithm adaptively determines the DA operator $\mathbf{A}_c$ and DA intensity $\mathbf{E}_c$ for each class based on information such as the accuracy of each class in the previous epoch and historical accuracy.

The key aspect of LLM-AutoDA lies in the synergy between the DA strategies generated by LLMs and the long-tailed learning model. This close interaction allows the discovered DA strategies to dynamically align with the model training process, effectively enhancing the performance of long-tailed learning in a targeted manner.

## 4.2 Strategy Evaluation

To evaluate the performance of candidate data augmentation strategies, we insert them into the model training process, conduct a small amount of additional training on the training set, and then test the accuracy on the validation set, using the accuracy as the fitness score for the algorithm. Assuming the data augmentation function generated by LLM is denoted as $f_{aug}$ and the training-testing function is denoted as $\mathcal{T}$, the evaluation process can be represented as follows:

$$Fitness(f_{aug}) = \mathcal{T}(f_{aug}, e_{ckp}, N_{ft}) \tag{1}$$

where $e_{ckp}$ is the starting checkpoint epoch number and $N_{ft}$ is the epoch number of fine-tune. The function $\mathcal{T}$ injects $f_{aug}$ into the training flow:

$$\mathcal{T}(f_{aug}, e_{ckp}, N_{ft}) = Acc_{val}(Fine-tune(f_{aug}, \mathcal{D}_{train}, e_{ckp}, N_{ft})) \tag{2}$$

The $Finetune$ function starts training from the $e_{ckp}$ checkpoint and performs $N_{ft}$ epochs of training on the training set $\mathcal{D}_{train}$. At the beginning of each epoch, it dynamically selects data augmentation methods for each class using $faug$.

$$\mathbf{A}_c^{(t)}, \mathbf{E}_c^{(t)} = f_{aug}(\mathbf{W}_c^{(t-1)}, \mathbf{a}_c^{(t-1)}, \mathbf{H}_c^{(t-1)}, \mathbf{A}_c^{(t-1)}, \mathbf{E}_c^{(t-1)}, t) \tag{3}$$

Here, $\mathbf{A}_c^{(t)}$ represents the DA selection matrix for class $c$ at time $t$, $\mathbf{E}_c^{(t)}$ represents the corresponding augmentation intensity, $\mathbf{W}_c^{(t-1)}$ represents the weights of each augmentation method on class $c$ from the previous time step, $\mathbf{a}_c^{(t-1)}$ represents the accuracy of class $c$ at time $t-1$, and $\mathbf{H}_c^{(t-1)}$ represents the historical accuracy of class $c$ when using different augmentation methods in the previous step.

After training, the model is evaluated using the no augmented validation set $\mathcal{D}_{val}$, and the overall accuracy $Acc_{val}$ is obtained as the fitness score for $f_{aug}$.

$$Fitness(f_{aug}) = Acc_{val}(Finetune(f_{aug}, \mathcal{D}_{train}, e_{ckp}, N_{ft}), \mathcal{D}_{val}) \tag{4}$$

A higher fitness score indicates better performance of the algorithm on long-tailed distributions. By injecting candidate algorithms into the real training process and evaluating them on the validation set, we can accurately and efficiently measure their actual ability to address the long-tailed problem.

## 4.3 LLM-based Search Operator

LLM-AutoDA leverages Pretrained Language Models (PLMs) to automatically generate data augmentation algorithms. To guide the PLM in generating algorithms that meet specific requirements, we employ prompt engineering techniques and carefully design a series of prompts. By incorporating task descriptions, input-output formats, novelty requirements, and other prior knowledge through prompts, we can constrain the generation process of the PLM within the desired search space. We design the following three types of search operators, corresponding to different prompt templates:

- **Initialization operator** $I$: Based on the task description prompt $P_{task}$ and the knowledge base of data augmentation $\mathcal{K}$, a set of randomly initialized population algorithms $A_i^{(0)}{}_{i=1}^N$ is generated.

- **Crossover operator** $E$: Building upon $P_{task}$, $N_p$ parent algorithms $A_i^{(t)}i = 1^{N_p}$ from the current population are used as references, along with the incorporation of knowledge base $\mathcal{K}$. The PLM is required to generate $N_e$ new algorithms $A_j^{(t)}j = 1^{N_e}$ that are different in both form and logic from the existing algorithms, thereby expanding the search space.

- **Mutation operator** $M$: Based on $P_{task}$, $N_m$ individuals $A_i^{(t)}{}_{i=1}^{N_m}$ are selected from the current population, and local improvement directions are provided. The PLM is tasked with generating a mutated algorithm $\hat{A}_i^{(t)}$ for each $A_i^{(t)}$ within its neighborhood for further exploration.

Taking the crossover operator $E$ as an example, its prompt $P_E$ can be represented as follows:

$$P_E(P_{task}, \{A_i^{(t)}\}_{i=1}^{N_p}, \mathcal{K}, D_{func}) = P_{task} + P_{ref}(\{A_i^{(t)}\}_{i=1}^{N_p}) + P_{know}(\mathcal{K}) + P_{diff} + P_{format}(D_{func}) \tag{5}$$

where $P_{task}$ represents the task description, $A_i i = 1^N$ represents $N$ parent algorithms, and $Dfunc$ represents the domain of the objective function. $P_{ref}$ formats the parent algorithms into reference code, $P_{diff}$ requires the generation of new algorithms that are completely different from the existing ones, and $P_{format}$ specifies the input and output of the objective function.

Once we have obtained the prompt $P_E$, we input it into the pretrained language model $\mathcal{L}$, and as a result, we obtain $N_e$ new crossover algorithms.

$$\{A_j^{(t)}\}_{j=1}^{N_e} = \mathcal{L}(P_E(P_{task}, \{A_i^{(t)}\}_{i=1}^{N_p}, \mathcal{K}, D_{func})) \tag{6}$$

Each $A_j^{(t)}$ typically consists of a natural language description of the algorithm and its corresponding Python code implementation. Similarly, the prompts $P_I$ and $P_M$ for the initialization operator $I$ and the mutation operator $M$ can be constructed in a similar manner, with the main difference lying in the introduced prior information.

# 5 Experiments

## 5.1 Experimental Settings

**Datasets and Metrics.** Like most long-tailed learning methods, we conducted experiments on several mainstream long-tailed learning datasets, including CIFAR-100-LT [5], ImageNet-LT [26], and iNaturalist 2018 [38]. Among them, CIFAR-100-LT is the long-tailed version of CIFAR-100, with various imbalance ratios. To validate the effectiveness of LLM-AutoDA in addressing the long-tailed problem, we selected three testing environments: 50, 100, 200. Compared to CIFAR-100-LT, both ImageNet-LT and iNaturalist 2018 have more classes and larger data sizes. It is worth noting that, similar to CIFAR-100-LT, ImageNet-LT is a long-tailed version artificially constructed from the well-known ImageNet [35] dataset. On the other hand, iNaturalist 2018 is a naturally occurring long-tailed dataset collected from the real world. We used the official complete versions of these datasets, and detailed information about the datasets is provided in Appendix B. We use Top-1 accuracy as the evaluation metric and provide the performance of subsets based on the class divisions provided by the official datasets.

**Baselines.** Following the settings of CUDA [2], we considered various research theories when selecting the baselines. In addition to the classic cross-entropy loss (CE) [15], we also validated different data augmentation methods on other baselines, such as loss-based re-balancing methods: CE-DRW [5], LDAM-DRW [5], Balanced Softmax (BS) [33], and model-based re-balancing methods: RIDE [43], BCL [48]. In terms of data augmentation methods, we compared LLM-AutoDA with the latest SOTA methods: CUDA [2] and DODA [39]. We observed their advantages and disadvantages by combining these DA methods with the long-tailed baselines.

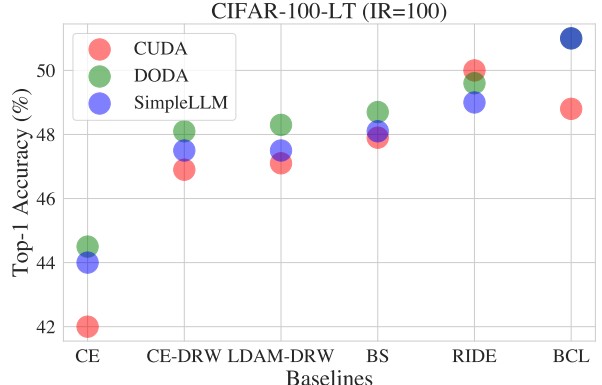

Figure 4: Average accuracy (%) on CIFAR-100-LT dataset (Imbalance ratio=100) with CUDA and DODA. SimpleLLM is comparable to the effectiveness of CUDA and DODA when combined with long-tailed learning baselines.

The relevant descriptions of the baselines are also provided in Appendix A.

**Implementation Details.** All our models are implemented based on PyTorch [27]. We trained and evaluated the models on 2 NVIDIA Tesla A100 GPUs and reported the experimental results. We utilized the powerful gpt-3.5-turbo for strategy generation and employed AEL [23] for strategy optimization. In the experimental process, we first trained the models for 50 epochs without using augmentation strategies, then continued training with augmentation strategies for an additional 20 epochs, employing a novel evaluation mechanism. Additionally, during the final evaluation stage of long-tailed learning, we adopted the same settings as DODA [39] for all baseline methods and our approach.

## 5.2 Comparison with the State-of-the-art

**Results on CIFAR-100-LT.** We first evaluated LLM-AutoDA and other long-tailed data augmentation (DA) methods on CIFAR-100-LT dataset (IR = 50, 100). The experimental results are shown in Table 1. From the results, it can be observed that both SimpleLLM and the improved version LLM-AutoDA significantly improve the global accuracy of the model compared to the original long-tailed learning baseline, achieving robust improvements. In the horizontal comparison with long-tailed DA methods CUDA [2] and DODA [39], as analyzed earlier, SimpleLLM achieves comparable performance to the former through a non-optimized way. This indicates that within our framework, a locally optimal strategy can replace a carefully designed complex strategy. In addition, LLM-AutoDA brings more significant and stable gains, reflecting that existing long-tailed DA methods may be suboptimal strategies within our augmentation strategy space, while LLM-AutoDA can provide the

Table 1: Accuracy (%) on CIFAR-100-LT dataset (Imbalance ratio={50, 100}) with SOTA DA methods. **Blod** indicates the best performance while underline indicates the second best. (+) and (-) indicate the relative gain.

| Method | IR = 50 | | | | IR = 100 | | | |
|---|---|---|---|---|---|---|---|---|
| | Head | Medium | Tail | All | Head | Medium | Tail | All |
| CE [15] | 63.9 | 36.2 | 15.2 | 43.8 (+0.0) | 65.6 | 36.2 | 8.2 | 40.1 (+0.0) |
| CE + CUDA | 68.3 | 38.4 | 13.7 | 46.2 (+2.4) | 70.7 | 41.4 | 9.3 | 42.0 (+3.9) |
| CE + DODA | 71.2 | 40.3 | 12.6 | 48.0 (+4.2) | 74.8 | 43.8 | 10.0 | 44.5 (+6.4) |
| CE + SimpleLLM | 71.4 | 39.9 | 13.1 | 48.0 (+4.2) | 72.5 | 44.9 | 9.8 | 44.0 (+5.9) |
| CE + LLM-AutoDA | **72.3** | 40.0 | 14.1 | 48.6 (+4.8) | **74.9** | 45.3 | 9.6 | 45.0 (+6.9) |
| CE-DRW [5] | 60.6 | 39.0 | 22.9 | 45.0 (+0.0) | 63.4 | 41.2 | 15.7 | 41.4 (+0.0) |
| CE-DRW + CUDA | 63.8 | 48.0 | 37.0 | 52.5 (+7.5) | 63.5 | 48.9 | 25.3 | 46.9 (+5.5) |
| CE-DRW + DODA | 63.4 | 47.4 | 38.9 | 52.5 (+7.5) | 60.2 | **51.9** | 29.6 | 48.1 (+6.7) |
| CE-DRW + SimpleLLM | 62.3 | 49.4 | 38.8 | 52.7 (+7.7) | 62.1 | 49.6 | 27.9 | 47.5 (+6.1) |
| CE-DRW + LLM-AutoDA | 63.1 | 48.4 | 39.3 | 52.8 (+7.8) | 62.9 | 50.7 | 29.9 | 48.7 (+7.3) |
| LDAM-DRW [5] | 63.0 | 41.2 | 25.1 | 47.2 (+0.0) | 62.8 | 42.6 | 21.1 | 43.2 (+0.0) |
| LDAM-DRW + CUDA | 66.2 | 46.2 | 26.4 | 50.8 (+3.6) | 66.0 | 49.5 | 22.1 | 47.1 (+3.9) |
| LDAM-DRW + DODA | 64.7 | 46.3 | 27.5 | 50.5 (+3.3) | 65.4 | 50.8 | 25.5 | 48.3 (+5.1) |
| LDAM-DRW + SimpleLLM | 65.1 | 45.2 | 27.3 | 50.1 (+2.9) | 65.7 | 49.4 | 23.9 | 47.5 (+4.3) |
| LDAM-DRW + LLM-AutoDA | 66.7 | 46.1 | 27.4 | 51.2 (+4.0) | 66.7 | 50.1 | 26.3 | 48.8 (+5.6) |
| BS [33] | 60.3 | 41.3 | 34.3 | 47.9 (+0.0) | 59.6 | 42.3 | 23.7 | 42.8 (+0.0) |
| BS + CUDA | 63.6 | 48.4 | 37.3 | 52.7 (+4.8) | 62.5 | 49.1 | 29.4 | 47.9 (+5.1) |
| BS + DODA | 62.2 | **51.2** | 41.5 | 54.0 (+6.1) | 63.1 | 49.3 | 31.2 | 48.7 (+5.9) |
| BS + SimpleLLM | 62.4 | 48.8 | 37.7 | 52.4 (+4.5) | 62.4 | 48.8 | 30.6 | 48.1 (+5.3) |
| BS + LLM-AutoDA | 63.3 | 50.5 | 40.2 | 53.9 (+6.0) | 63.3 | 50.0 | 31.0 | 49.0 (+6.2) |
| RIDE [43] | 65.7 | 47.7 | 31.8 | 52.2 (+0.0) | 65.7 | 48.6 | 25.0 | 47.5 (+0.0) |
| RIDE + CUDA | 67.8 | 47.0 | 33.4 | 53.1 (+0.9) | 67.9 | 51.2 | 27.6 | 50.0 (+2.5) |
| RIDE + DODA | 68.2 | 46.1 | 29.3 | 52.1 (-0.1) | 68.7 | 50.9 | 25.7 | 49.6 (+2.1) |
| RIDE + SimpleLLM | 67.3 | 46.8 | 30.8 | 52.3 (+0.1) | 69.3 | 48.8 | 25.4 | 49.0 (+1.5) |
| RIDE + LLM-AutoDA | 67.1 | 47.3 | 32.7 | 52.8 (+0.6) | 69.1 | 50.2 | 28.1 | 50.2 (+2.7) |
| BCL [48] | 61.6 | 43.1 | 34.3 | 49.1 (+0.0) | 63.1 | 42.9 | 23.9 | 44.2 (+0.0) |
| BCL + CUDA | 64.0 | 47.4 | 39.4 | 52.7 (+3.6) | 64.7 | 49.7 | 29.1 | 48.8 (+4.6) |
| BCL + DODA | 64.9 | 48.0 | 40.6 | 53.6 (+4.5) | 66.0 | 50.7 | **33.8** | 51.0 (+6.8) |
| BCL + SimpleLLM | 65.0 | 49.2 | 39.9 | 54.0 (+4.9) | 64.1 | 50.4 | 30.1 | 49.0 (+4.8) |
| BCL + LLM-AutoDA | 64.9 | 49.2 | **44.1** | **54.8** (+5.7) | 66.6 | 50.6 | 33.1 | 51.0 (+6.8) |

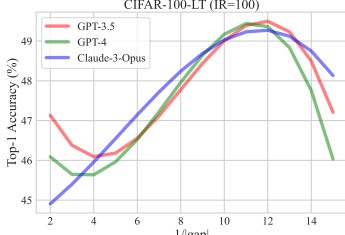
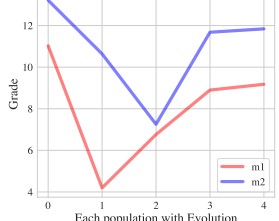
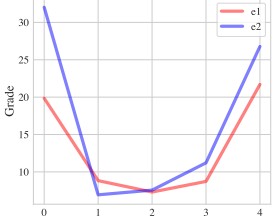

Figure 5: Impact of different LLMs on the performance of long-tailed learning models.

Figure 6: Impact of different population numbers in the mutation prompts.

Figure 7: Impact of different population numbers in the crossover prompts.

optimal augmentation strategy through continuous optimization. To evaluate the effectiveness of LLM-AutoDA in highly imbalanced scenarios, we adjusted the imbalance ratio to 200 and conducted comparative experiments in Appendix D.

**Results on ImageNet-LT and iNaturalist 2018.** We also conducted comparative experiments on large-scale benchmark datasets, ImageNet-LT and iNaturalist 2018. As expected, different long-tailed learning methods showed significant performance improvement when integrated with LLM-AutoDA. Similar to the highly imbalanced setting mentioned earlier, both of these large-scale datasets are inherently highly imbalanced. Therefore, the augmentation strategies provided by LLM-AutoDA can consistently demonstrate superiority in various imbalanced environments. Importantly, LLM-AutoDA does not rely on meticulous manual design, which reduces the optimization cost on large-scale datasets.

Table 2: Accuracy (%) on ImageNet-LT and iNaturalist 2018 datasets with SOTA DA methods. **Blod** indicates the best performance while underline indicates the second best. (+) and (-) indicate the relative gain.

| Method | ImageNet-LT | | | | iNaturalist 2018 | | | |
|---|---|---|---|---|---|---|---|---|
| | Head | Medium | Tail | All | Head | Medium | Tail | All |
| CE [15] | 64.0 | 33.8 | 5.8 | 41.6 (+0.0) | 73.9 | 63.5 | 55.5 | 61.0 (+0.0) |
| CE + CUDA | 67.1 | 47.1 | 13.4 | 47.2 (+5.6) | 74.6 | 65.0 | 57.2 | 62.5 (+1.5) |
| CE + DODA | 67.4 | 47.5 | 13.9 | 48.1 (+6.5) | 74.9 | 66.0 | 58.4 | 63.6 (+2.6) |
| CE + LLM-AutoDA | 68.2 | 47.1 | 14.3 | 50.4 (+8.8) | 75.1 | 66.3 | 58.9 | 64.0 (+3.0) |
| CE-DRW [5] | 61.7 | 47.3 | 28.8 | 50.1 (+0.0) | 68.2 | 67.3 | 66.4 | 67.0 (+0.0) |
| CE-DRW + CUDA | 61.7 | 48.4 | 30.5 | 51.1 (+1.0) | 68.8 | 67.9 | 66.5 | 67.4 (+0.4) |
| CE-DRW + DODA | 62.4 | 48.5 | 31.3 | 52.2 (+2.1) | 69.0 | 68.2 | 67.8 | 68.2 (+1.2) |
| CE-DRW + LLM-AutoDA | 62.8 | 48.3 | 31.7 | 51.6 (+1.5) | 68.8 | 68.8 | 68.1 | 68.7 (+1.7) |
| LDAM-DRW [5] | 60.4 | 46.9 | 30.7 | 49.8 (+0.0) | - | - | - | 66.1 (+0.0) |
| LDAM-DRW + CUDA | 63.2 | 48.2 | 31.2 | 51.5 (+1.7) | 68.0 | 67.5 | 66.8 | 67.3 (+1.2) |
| LDAM-DRW + DODA | 63.7 | 48.6 | 31.9 | 52.4 (+2.6) | 68.6 | 68.1 | 67.9 | 68.7 (+2.6) |
| LDAM-DRW + LLM-AutoDA | 63.3 | 49.4 | 32.4 | 52.5 (+2.7) | 68.0 | 69.4 | 68.6 | 69.5 (+3.4) |
| BS [33] | 60.9 | 48.8 | 32.1 | 51.0 (+0.0) | 65.7 | 67.4 | 67.5 | 67.3 (+0.0) |
| BS + CUDA | 61.8 | 49.1 | 31.8 | 51.5 (+0.5) | 67.6 | 68.2 | 68.3 | 68.2 (+0.9) |
| BS + DODA | 61.9 | 49.5 | 32.4 | 52.0 (+1.0) | 68.1 | 68.9 | 69.5 | 69.4 (+2.1) |
| BS + LLM-AutoDA | 62.5 | 50.0 | 32.8 | 52.5 (+1.5) | 68.0 | 69.1 | 69.9 | 69.8 (+2.5) |
| RIDE [43] | 64.9 | 50.4 | 34.4 | 53.6 (+0.0) | 70.4 | 71.8 | 71.8 | 71.6 (+0.0) |
| RIDE + CUDA | 66.0 | 51.7 | 34.7 | 54.7 (+1.1) | 70.6 | 72.6 | 72.7 | 72.4 (+1.4) |
| RIDE + DODA | 66.6 | 51.9 | 35.9 | 55.8 (+2.2) | 70.9 | 72.4 | 73.9 | 73.7 (+2.8) |
| RIDE + LLM-AutoDA | 67.1 | 52.3 | 37.3 | 56.5 (+2.9) | 70.9 | 72.8 | 73.8 | 73.9 (+3.0) |
| BCL [48] | 65.3 | 53.5 | 36.3 | 55.6 (+0.0) | 69.4 | 72.4 | 71.8 | 71.8 (+0.0) |
| BCL + CUDA | 66.8 | 53.9 | 36.6 | 56.3 (+0.7) | 70.8 | 72.7 | 72.0 | 72.2 (+0.4) |
| BCL + DODA | 66.9 | 54.1 | 37.4 | 56.9 (+1.3) | 71.2 | 73.2 | 73.4 | 73.7 (+1.9) |
| BCL + LLM-AutoDA | 67.2 | 55.1 | 38.3 | 57.5 (+1.9) | 70.9 | 73.6 | 74.7 | 74.2 (+2.4) |

## 5.3 More Analysis and Discussion

**Do different LLMs produce differentiated effects?** In the aforementioned experiments, we used GPT-3.5 [3] as the LLM to respond to the designed prompts. To analyze whether LLM-AutoDA is dependent on specific LLM models (e.g., GPT-3.5), we replaced the LLM model in LLM-AutoDA with other popular methods such as GPT-4 [1] and Claude-3-Opus, and conducted experiments. The experimental results, shown in Figure 5, demonstrate that all three LLMs exhibit consistent performance trends when selecting augmentation strategies in different score ranges. For instance, they all show high performance near the augmentation strategies with scores around 12, while augmentation strategies with excessively high scores lead to performance degradation across the three LLMs due to insufficient diversity.

**Do different population numbers have an impact on performance?** In LLM-AutoDA, we employed two different types of prompts: crossover prompts and mutation prompts. Crossover prompts involve transforming multiple parent populations into a single population, while mutation prompts replace the current augmentation strategy with an equivalent one.

Throughout the iterative process of framework evolution, when performing crossover and mutation operations, we need to specify the number of populations generated each time. Figures 6 and 7 illustrate the scores of strategies generated by two different mutation prompts (m1, m2) and two different crossover prompts (e1, e2), respectively. It can be observed that different prompts exhibit consistent trends in score variations. Particularly, compared to e1, e2 demonstrates a higher score variance, which indirectly reflects the bias in its prompt content.

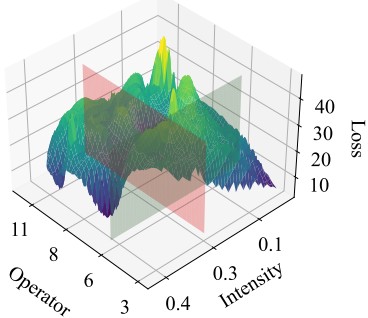

Figure 8: Visualization of the process of finding the optimal solution for different augmentation paradigms.

**Why are fixed strategy methods often local optima?** Strategy fixed data augmentation methods, such as CUDA and DODA, aim to adapt to long-tailed distributions by dynamically adjusting the augmentation operators or intensities. However, their focus is limited. We visualized the loss variations when searching for optimal augmentation strategies using different paradigms. From Figure 8, it can be observed that CUDA (i.e., green plane) and DODA (i.e., red

plane) can only search for local optimal strategies on a single plane, while LLM-AutoDA is capable of flexibly searching for the points with the lowest loss across the entire curved surface to obtain a global optimal solution.

## 6 Conclusion

Existing paradigms for long-tailed learning have partially alleviated the long-tailed problem but still have limitations. To address this, this paper presents an LLM-driven long-tailed data augmentation framework called LLM-AutoDA, which utilizes large-scale pre-trained language models to automatically search for data augmentation strategies optimized for long-tailed data distributions. Experiments on multiple mainstream benchmark datasets demonstrate that LLM-AutoDA outperforms state-of-the-art methods.

## Acknowledgements

This work was supported by the Natural Science Foundation of China Youth Project (No. 62402472), the Natural Science Foundation of Jiangsu Province of China Youth Project (No. BK20240461), the Research Grants Council of the Hong Kong Special Administrative Region, China (GRF Project No. CityU 11215723), National Natural Science Foundation of China (No.62072427, No.12227901), the Project of Stable Support for Youth Team in Basic Research Field, CAS (No.YSBR-005), and Academic Leaders Cultivation Program, USTC.

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

# Appendix
# LLM-AutoDA: Large Language Model-Driven Automatic Data Augmentation for Long-tailed Problems

The content of the **Appendix** is summarized as follows:

## A  Baselines Details

**Cross-Entropy Loss (CE) [15]**   As a classic classification loss function, cross-entropy (CE) is widely used in both balanced and imbalanced data distributions, directly computing the loss based on the true labels of samples and the predicted probability distributions of the model. Although simple and effective, CE tends to overly focus on head classes while overlooking tail classes in long-tailed scenarios.

**Loss-based Re-balancing Strategies**

- **CE-DRW** [5] combines re-weighting of training samples based on the inverse class frequency and gradient reversal that considers class frequencies during backpropagation, enhancing the focus on tail classes.
- **LDAM-DRW** [5] further improves upon CE-DRW by introducing learnable amplification factors to automatically adjust the weight of each class, better optimizing the inter-class balance.
- **Balanced Softmax (BS)** [33] re-weights the logits based on the prior class probabilities, giving higher attention to tail classes.

**Model-based Re-balancing Strategies**

- **RIDE** [43] ensembles multiple expert models, each focusing on different class distributions, and adaptively combines their outputs based on the test data to adapt to distribution shifts.
- **BCL** [48] for CNN classifiers, optimizes the losses of head and tail components while considering intra-class variance and inter-class distances to enhance the separability of classification boundaries.

**Data Augmentation Methods**

- **CUDA** [2] proposes a contrastive learning-based automatic data augmentation method that generates transformations during training, significantly boosting performance in long-tailed scenarios.
- **DODA** [39] dynamically adjusts the data augmentation strategies for different classes from a distribution perspective, ensuring sufficient augmentation for tail classes while avoiding over-augmentation for head classes.

The above baseline methods offer unique insights into addressing long-tailed distributions, laying the foundation for our research. The experimental section will comprehensively compare and analyze the performance of these methods across different datasets and evaluation metrics.

## B  Evaluation Datasets

To comprehensively evaluate the effectiveness of our proposed method, we conduct experiments on four representative long-tailed datasets: CIFAR100-LT, ImageNet-LT, iNaturalist 2018, and Places365-LT. These datasets cover diverse domains and exhibit varying degrees of class imbalance, providing a comprehensive and challenging testbed for long-tailed learning algorithms.

Table 3: Statistics of the long-tailed datasets.

| Dataset | # Classes | # Train | # Test | Max Imbalance Ratio |
|---------|-----------|---------|--------|---------------------|
| CIFAR100-LT | 100 | 50,000 | 10,000 | 100 |
| ImageNet-LT | 1,000 | 115,846 | 50,000 | 256 |
| iNaturalist 2018 | 8,142 | 437,513 | 24,426 | 500 |

**CIFAR100-LT [5]** is the long-tailed version of the renowned CIFAR100 dataset, comprising 60,000 $32 \times 32$ color images across 100 classes. The long-tailed distribution is induced by exponentially decreasing the number of samples per class, with a maximum imbalance ratio of 100.

**ImageNet-LT [26]** is a long-tailed subset of the large-scale ImageNet dataset, containing over 115,000 images spanning 1,000 classes. The class cardinalities follow a Pareto distribution with $\alpha = 6$, leading to a maximum imbalance ratio of 256.

**iNaturalist 2018 [38]** is a real-world dataset reflecting the long-tailed distribution in nature, comprising approximately 450,000 images across 8,142 species categories. Due to the drastic variation in the number of images per species, this dataset has a maximum imbalance ratio of 500, posing a significant challenge with extreme class imbalance and high intra-class variation.

## C  Pseudocode

---

**Algorithm 1:** LLM-AutoDA: Automatic Data Augmentation with Language Models

---

1: **Input:**
2: - Long-tailed datasets $\mathcal{D}_{train}, \mathcal{D}_{val}$
3: - Pretrained language model $\mathcal{L}$
4: - Initial data augmentation policies $\mathcal{K}$
5: - Long-tailed learning model $f_\theta$ with parameters $\theta$
6: **Output:**
7: - Optimal data augmentation algorithm $\mathcal{A}^*$
8: - Final model $f_\theta$
9: Define task description prompt $\mathcal{P}_{task}$
10: Define exploration operator prompt template $\mathcal{P}_E$
11: Define mutation operator prompt template $\mathcal{P}_M$
12: Initialize algorithm population $\{\mathcal{A}_i^{(0)}\}_{i=1}^N = \text{INITIALIZE}(\mathcal{L}, \mathcal{P}_{task}, \mathcal{K})$
13: **for** each generation $t$ **do**
14:    $\{\mathcal{A}_i^{(t)}\}_{i=1}^{N_p} = \text{SELECT}(\{\mathcal{A}_i^{(t-1)}\}_{i=1}^N)$ {Select parents}
15:    $\{\mathcal{A}_j^{(t)}\}_{j=1}^{N_e} = \text{EXPLORE}(\mathcal{L}, \mathcal{P}_E, \{\mathcal{A}_i^{(t)}\}_{i=1}^{N_p}, \mathcal{K})$ {Explore}
16:    $\{\hat{\mathcal{A}}_i^{(t)}\}_{i=1}^{N_m} = \text{MUTATE}(\mathcal{L}, \mathcal{P}_M, \{\mathcal{A}_i^{(t)}\}_{i=1}^{N_m}, \mathcal{K})$ {Mutate}
17:    $\{\mathcal{A}_i^{(t)}\}_{i=1}^N = \{\mathcal{A}_i^{(t)}\}_{i=1}^{N_p} \cup \{\mathcal{A}_j^{(t)}\}_{j=1}^{N_e} \cup \{\hat{\mathcal{A}}_i^{(t)}\}_{i=1}^{N_m}$
18:    **for** each $\mathcal{A}_i^{(t)}$ **do**
19:       $fitness_i^{(t)} = \text{EVALUATE}(\mathcal{A}_i^{(t)}, f_\theta, \mathcal{D}_{train}, \mathcal{D}_{val}, e_{ckp}, N_{ft})$
20:    **end for**
21: **end for**
22: $\mathcal{A}^* =_{\mathcal{A}_i} fitness_i$ {Select algorithm with highest fitness}
23: **return** $\mathcal{A}^*, f_\theta$
   =0

---

where INITIALIZE uses an initialization operator to generate a random population of algorithms, EXPLORE and MUTATE correspond to the exploration and mutation operators respectively. In each generation, a subset of individuals is selected from the previous generation as parents, then new algorithms are generated using the exploration and mutation operators, and merged into the population. For each candidate algorithm, EVALUATE is called to compute its fitness:

```
function EVALUATE($\mathcal{A}, f_\theta, \mathcal{D}_{train}, \mathcal{D}_{val}, e_{ckp}, N_{ft}$):
    $f'_\theta$ = FINETUNE($\mathcal{A}, f_\theta, \mathcal{D}_{train}, e_{ckp}, N_{ft}$)
    return ACCURACY($f'_\theta, \mathcal{D}_{val}$) =0
```

FINETUNE starts from the $e_{ckp}$-th checkpoint and incrementally trains for $N_{ft}$ epochs on the training set, with the algorithm $\mathcal{A}$ dynamically selecting augmentations for each class:

```
function FINETUNE($\mathcal{A}, f_\theta, \mathcal{D}_{train}, e_{ckp}, N_{ft}$):
    Load checkpoint of $f_\theta$ at epoch $e_{ckp}$
    for $e = e_{ckp}$ to $e_{ckp} + N_{ft}$ do
        for each class $c$ do
            $\mathcal{A}_c^{(e)}, \mathcal{E}_c^{(e)} = \mathcal{A}(\mathbf{W}_c^{(e-1)}, \mathbf{H}_c^{(e-1)}, acc_c^{(e-1)}, \mathcal{A}_c^{(e-1)}, \mathcal{E}_c^{(e-1)}, e)$
        end for
        Train $f_\theta$ for one epoch on $\mathcal{D}_{train}$ using $\{\mathcal{A}_c^{(e)}, \mathcal{E}_c^{(e)}\}$ for augmentation
    end for
    return $f_\theta$ =0
```

where $\mathbf{W}_c^{(e-1)}$ are the weights of augmentation techniques for class $c$, $\mathbf{H}_c^{(e-1)}$ is the history of accuracies for $c$, and $acc_c^{(e-1)}$ is the accuracy in the previous epoch.

## D  Future Anylysis

### D.1  Highly Imbalanced Scenarios

To evaluate the effectiveness of LLM-AutoDA in highly imbalanced scenarios, we adjusted the imbalance ratio to 200 and conducted comparative experiments. The experimental results, as shown in Figure 9, demonstrate that LLM-AutoDA consistently outperforms other long-tailed data augmentation methods.

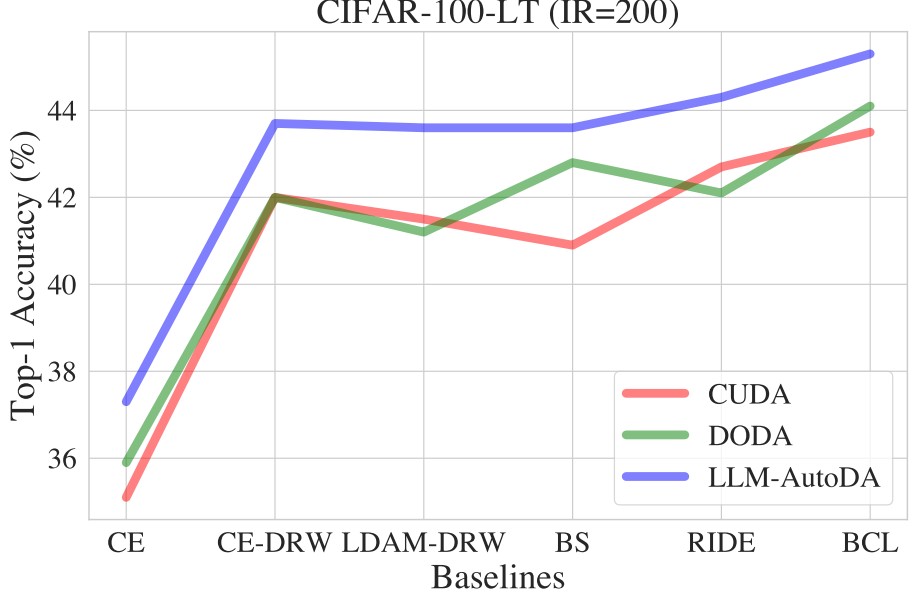

Figure 9: Accuracy (%) on more imbalanced CIFAR-100-LT dataset (Imbalance ratio=200) with SOTA DA methods.

## D.2 Visualization of Selection Process

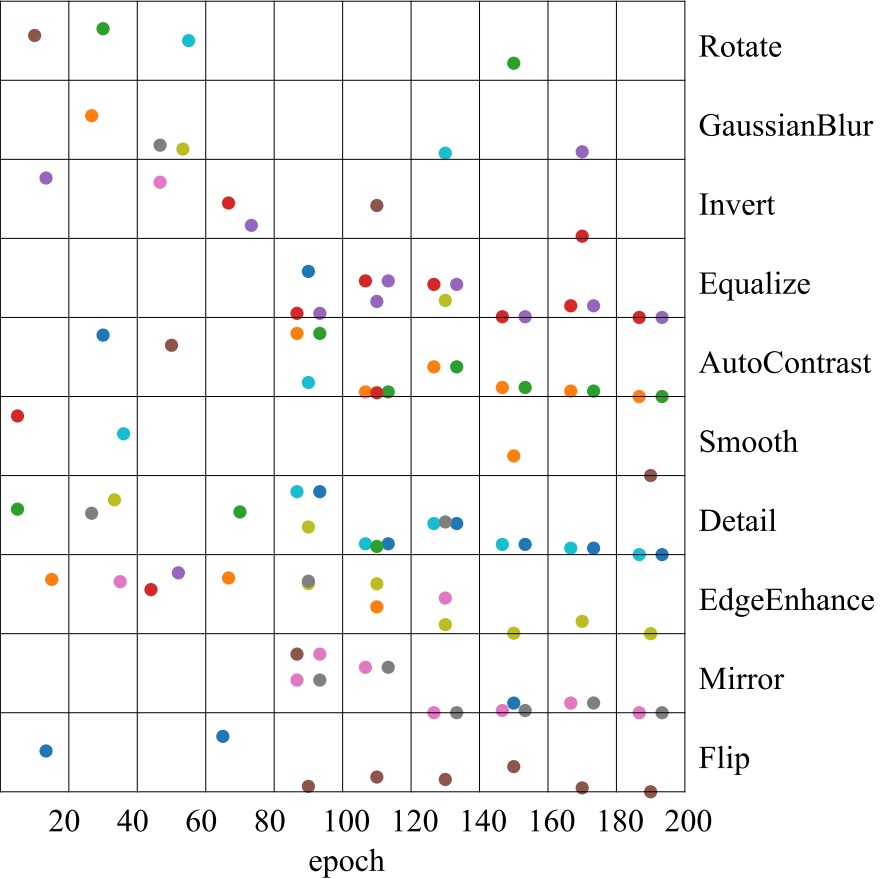

Figure 10: Trends in the augmentation intensities and number of times different strategies are selected.

In the figure13, we visualize the selection process of the data augmentation strategies provided by the model. We train the discovered data augmentation methods on CIFAR-100 with an imbalance ratio of 100 under the bi-entropy loss, where the x-axis is the epoch. The y-axis represents the data augmentation techniques that may be used in this work, including Mirror, EdgeEnhance, Detail, Smooth, AutoContrast, Equalize, Invert, GaussianBlur, Rotate, and Flip. The augmentation intensity ranges from 0 to 1 for all of these techniques. For samples of the first class, on the right side, different data augmentation methods are represented, with a dot in the grid indicating that the method represented by that row is selected, and dots of different colors indicating multiple selections. The height in the grid represents the intensity given by our strategy. In the early stages of training, the selected data augmentation strategies are relatively random, but as the model continues training, the trends in the selected data augmentation methods, intensities, and number of times selected gradually stabilize. This shows that the data augmentation methods provided by the large model can effectively achieve convergence on the long-tailed learning model. This demonstrates the feasibility of using a large model to search for and design data augmentation strategies for long-tailed learning models.

## D.3 Cost Analysis

Using LLM-AutoDA to optimize data augmentation strategies for long-tailed recognition tasks requires significantly less time compared to the manual design of data augmentation strategies by humans. LLM-AutoDA can obtain highly effective strategies within a mere 2 3 hours under a given framework.

## E Prompts

You are requested to design a novel algorithm that, given a set of augmentation techniques, selects several of them based on the change in per-class accuracy between the current training time point and the previous one, to be employed in the subsequent training phase, with the aim of enhancing the model's ability to tackle long-tail problems. This algorithm should deviate from existing methodologies present in the literature.

Describe the new algorithm and main steps in one sentence, place the sentence inside curly braces, Next, implement it in Python as a function named get_aug_type. This function should accept 6 input(s): 'aug_weight', 'ACCs', 'History_ACCs', 'lats_chose_matix', 'lats_chose_exts', 'epoch'. The function should return 1 output(s): 'chose_matrix,chose_exts'. aug_weight is a two-dimensional integer array initialized to 1, used to record the historical weight information for each category (indexed by rows) across every augmentation technique (indexed by columns). ACCs is a one-dimensional integer array showcasing the performance of each category at the current training instant, specifically, the count of correct predictions within that category. History_ACCs is a two-dimensional integer array that records the number of correct predictions made by each augmentation technique (column-wise) the last time they were employed for every category. lats_chose_matrix is a two-dimensional Boolean array indicating whether specific augmentation techniques (by column index) were utilized for each category (row index) in the previous training step; True signifies usage, while False denotes non-usage. lats_chose_exts is a two-dimensional float array representing the application intensity of each augmentation method across classifications at the current point in time, with a range from 0 to 1, where higher numbers imply greater enhancement strength. epoch is an integer denoting the current training epoch, indicating the number of completed training cycles. chose_matrix is a two-dimensional Boolean array marking which augmentation techniques (by column index) will be employed in the next training step; True values indicate adoption, and False, rejection. chose_exts is a two-dimensional float array signifying the intensity of applying each augmentation technique to individual categories in the upcoming time step, also ranging from 0 to 1, with larger values indicating more substantial augmentation efforts. All are Numpy arrays.

Do not give additional explanations.

Figure 11: An example of initialization prompts.

You are requested to design a novel algorithm that, given a set of augmentation techniques, selects several of them based on the change in per-class accuracy between the current training time point and the previous one, to be employed in the subsequent training phase, with the aim of enhancing the model's ability to tackle long-tail problems. This algorithm should deviate from existing methodologies present in the literature.

I have 1 existing algorithms with their codes as follows:

No.1 algorithm and the corresponding code are:

The algorithm dynamically adjusts augmentation technique selection and intensity based on per-class accuracy improvements, historical usage, and a decay factor to address class imbalance and enhance model performance on underrepresented classes.

```
import numpy as np
import random

def get_aug_type(aug_weight,ACCs, History_ACCs, lats_chose_matix, lats_chose_exts,epoch):
    cls_num,num_aug_type = History_ACCs.shape
    # solve a weight as self.aug_weight

    for cidx in range(cls_num):
        indices = lats_chose_matix[cidx]
        assert indices.any() ,f'class index {cidx} has no chose_aug (num of aug must > 0)
        aug_weight[cidx][indices] = np.where(ACCs[cidx] > History_ACCs[cidx][indices],
                    aug_weight[cidx][indices] + 1,
                    aug_weight[cidx][indices] - 1)
        aug_weight = np.maximum(aug_weight, 1)

    chose_aug = np.zeros((cls_num ,num_aug_type)).astype(bool)

    chose_exts = np.random.rand(*lats_chose_exts.shape)

    aug_list = [i for i in range(num_aug_type)]

    for i in range(cls_num):
        indexes = random.choices(aug_list , weights = aug_weight[i , : ].tolist() , k = 1) #self.args.MAX_N
        for index in indexes:
            chose_aug[i][index] = True
    return chose_aug,chose_exts
```

Please help me create a new algorithm that has a totally different form from the given ones.

Describe the new algorithm and main steps in one sentence, place the sentence inside curly braces. Next, implement it in Python as a function named get_aug_type. This function should accept 6 input(s): 'aug_weight', 'ACCs', 'History_ACCs', 'lats_chose_matix', 'lats_chose_exts', 'epoch'. The function should return 1 output(s): 'chose_matrix,chose_exts'. aug_weight is a two-dimensional integer array initialized to 1, used to record the historical weight information for each category (indexed by rows) across every augmentation technique (indexed by columns). ACCs is a one-dimensional integer array showcasing the performance of each category at the current training instant, specifically, the count of correct predictions within that category. History_ACCs is a two-dimensional integer array that records the number of correct predictions made by each augmentation technique (column-wise) the last time they were employed for every category. lats_chose_matix is a two-dimensional Boolean array indicating whether specific augmentation techniques (by column index) were utilized for each category (row index) in the previous training step; True signifies usage, while False denotes non-usage. lats_chose_exts is a two-dimensional float array representing the application intensity of each augmentation method across classifications at the current point in time, with a range from 0 to 1, where higher numbers imply greater enhancement strength. epoch is an integer denoting the current training epoch, indicating the number of completed training cycles. chose_matrix is a two-dimensional Boolean array marking which augmentation techniques (by column index) will be employed in the next training step; True values indicate adoption, and False, rejection. chose_exts is a two-dimensional float array signifying the intensity of applying each augmentation technique to individual categories in the upcoming time step, also ranging from 0 to 1, with larger values indicating more substantial augmentation efforts. All are Numpy arrays.

Do not give additional explanations.

Figure 12: An example of crossover prompts.

You are requested to design a novel algorithm that, given a set of augmentation techniques, selects several of them based on the change in per-class accuracy between the current training time point and the previous one, to be employed in the subsequent training phase, with the aim of enhancing the model's ability to tackle long-tail problems. This algorithm should deviate from existing methodologies present in the literature.

I have one algorithm with its code as follows. Algorithm description: The algorithm dynamically adjusts augmentation technique selection and intensity based on per-class accuracy improvements, historical usage, and a decay factor to address class imbalance and enhance model performance on underrepresented classes.

Code:
```
import numpy as np
import random

def get_aug_type(aug_weight,ACCs, History_ACCs, lats_chose_matix, lats_chose_exts,epoch):

    cls_num,num_aug_type = History_ACCs.shape

    # solve a weight as self.aug_weight
    for cidx in range(cls_num):
        indices = lats_chose_matix[cidx]
        assert indices.any() ,f'class index {cidx} has no chose_aug (num of aug must > 0)'
        aug_weight[cidx][indices] = np.where(ACCs[cidx] > History_ACCs[cidx][indices],
                    aug_weight[cidx][indices] + 1,
                    aug_weight[cidx][indices] - 1)
        aug_weight = np.maximum(aug_weight, 1)

    chose_aug = np.zeros((cls_num ,num_aug_type)).astype(bool)
    chose_exts = np.random.rand(*lats_chose_exts.shape)
    aug_list = [i for i in range(num_aug_type)]

    for i in range(cls_num):
        indexes = random.choices(aug_list , weights = aug_weight[i , : ].tolist() , k = 1) #self.args.MAX_N
        for index in indexes:
            chose_aug[i][index] = True
    return chose_aug,chose_exts
```

Please assist me in creating a new algorithm that has a different form but can be a modified version of the algorithm provided.

First, describe the new algorithm and main steps in one sentence, place the sentence inside curly braces.Next, implement it in Python as a function named get_aug_type. This function should accept 6 input(s): 'aug_weight', 'ACCs', 'History_ACCs', 'lats_chose_matix', 'lats_chose_exts', 'epoch'. The function should return 1 output(s): 'chose_matrix,chose_exts'. aug_weight is a two-dimensional integer array initialized to 1, used to record the historical weight information for each category (indexed by rows) across every augmentation technique (indexed by columns). ACCs is a one-dimensional integer array showcasing the performance of each category at the current training instant, specifically, the count of correct predictions within that category. History_ACCs is a two-dimensional integer array that records the number of correct predictions made by each augmentation technique (column-wise) the last time they were employed for every category. lats_chose_matix is a two-dimensional Boolean array indicating whether specific augmentation techniques (by column index) were utilized for each category (row index) in the previous training step; True signifies usage, while False denotes non-usage. lats_chose_exts is a two-dimensional float array representing the application intensity of each augmentation method across classifications at the current point in time, with a range from 0 to 1, where higher numbers imply greater enhancement strength. epoch is an integer denoting the current training epoch, indicating the number of completed training cycles. chose_matrix is a two-dimensional Boolean array marking which augmentation techniques (by column index) will be employed in the next training step; True values indicate adoption, and False, rejection. chose_exts is a two-dimensional float array signifying the intensity of applying each augmentation technique to individual categories in the upcoming time step, also ranging from 0 to 1, with larger values indicating more substantial augmentation efforts. All are Numpy arrays.

Do not give additional explanations.

Figure 13: An example of mutation prompts.

# F   Limitations

This paper aims to innovate the data augmentation paradigm in long-tailed learning, which greatly improves the degree of freedom of long-tailed data augmentation. However, there are still some limitations of our approach. For example, the scoring mechanism for augmentation strategies is not perfect, and a more comprehensive scoring mechanism is needed. In addition, how to break through the search space limitation and generate novel augmentation methods is also a problem to be solved.

# G   Broader Impacts

Traditional data augmentation is not the best choice for long-tail learning, and recent long-tail data augmentation methods still lack degrees of freedom. The positive impact of our method is that it can give a large number of augmentation strategies suitable for long-tail learning in a short time, which greatly reduces the time and human cost consumed to design the strategy. Of course, there is also a small negative impact, that is, it is easy to cause dependence on LLMs.

