# OpenReview forum: "LLM-AutoDA: Large Language Model-Driven Automatic Data Augmentation for Long-tailed Problems"
_NeurIPS.cc/2024/Conference — NeurIPS 2024 poster_

### Official Review · Reviewer_2Q2t · 2024-07-11

**Soundness:** 3
**Presentation:** 3
**Contribution:** 4
**Rating:** 8
**Confidence:** 4

**Summary:**

The paper presents a novel approach by ingeniously leveraging large language models (LLMs) to generate data augmentation strategies for long-tailed problems. Given the recent rapid advancements in LLMs and their potential applications beyond natural language processing, this integration of LLMs with long-tailed learning is both creative and timely.

**Strengths:**

1) The proposed LLM-AutoDA framework generates augmentation strategies and incorporates a feedback loop mechanism to optimize these strategies iteratively. The idea is novel and interesting.

2) The paper offers in-depth analysis and insights into the discovered augmentation strategies, which not only aids in understanding the model's working principles but also provides valuable guidance for future research in long-tailed learning.  From a practical perspective, this method significantly reduces the cost of manually designing augmentation strategies, which has important practical value for researchers and practitioners dealing with imbalanced datasets.

3) The paper presents extensive experiments on multiple mainstream long-tailed learning benchmarks, convincingly demonstrating the method's effectiveness across various datasets and imbalance ratios.

**Weaknesses:**

1) The paper proposes multiple prompts to collaborate to update the data augmentation strategy, please explain what impact they will have when multiple prompts collaborate.

2) Can you try to evaluate and compare the performance of SimpleLLM and LLM-AutoDA under different long-tailed settings such as different imbalance ratios? I think this better reflects the ability of the searched strategy to adapt to different long-tailed scenarios.

3) This paper has some syntax errors, please improve.

**Questions:**

It is interesting that the authors upgrade the data augmentation paradigm of long-tailed learning, I wonder if this paradigm can also be combined with other domains with interesting impact. If the author can give relevant analysis, it will help to improve the audience of the paper.

**Limitations:**

Yes

---

> ### Author Rebuttal · Authors · 2024-08-07
>
> Thank you very much for your valuable comments and questions. Let me respond to them one by one:
>
> **W1:** In our framework, different prompts play complementary roles. For example, some prompts are responsible for generating new augmentation algorithms, while others mutate existing algorithms. Through this collaboration, we can strike a balance between exploring new augmentation spaces and leveraging known effective algorithms.
>
> The collaboration of multiple prompts helps the search strategy more comprehensively cover the design space of data augmentation. Different prompts can optimize the augmentation strategy from different perspectives, jointly promoting performance improvement. We will supplement more analysis in the paper to explain the benefits of this collaborative mechanism.
>
> **W2:** You made a great suggestion. Comparing the performance of the two methods under different imbalance ratios can more comprehensively evaluate the ability of the searched strategies to adapt to different long-tailed scenarios.
>
> In the revised version, we will add more experiments with different imbalance ratios to systematically analyze the strengths and weaknesses of SimpleLLM and LLM-AutoDA in dealing with different long-tail distributions. This will provide readers with richer experimental results and demonstrate the robustness of our method.
>
> **W3:** Thank you for the reminder. We will carefully check the entire text again, correct the grammatical errors found, and improve the expression quality of the paper.
>
> **Q1:** You raised an interesting direction for reflection. We believe that the paradigm of using LLM to assist data augmentation has the potential to be extended to other tasks and bring inspiration to more fields.
>
> For example, in natural language processing, LLM can generate targeted augmentation strategies based on the characteristics of text data, such as synonym replacement and back-translation, to alleviate data sparsity problems. In the field of speech recognition, LLM may also help optimize audio augmentation and improve the generalization of models.
>
> In the revised version, we will add some extensibility analysis in the discussion section to explore the possibilities and potential impacts of combining this paradigm with other fields.
>
> Thank you again for your feedback, which is very helpful for improving our work. We will carefully revise the paper, hoping that the revised version can answer your doubts and meet the publication standards.

---

> > ### Comment · Reviewer_2Q2t · 2024-08-12
> >
> > I appreciate the authors' response, which has largely addressed my concerns. Furthermore, I noticed that in the public response and in the reply to reviewer 37A8, the authors provided additional information about the evolutionary process and related strategies. I commend the efforts of both the reviewers and the authors. I believe that the demonstration of the process and the insights revealed highlight the effectiveness of large language models in addressing the challenges of long-tailed data distributions within this framework. This further convinces me that this paper will bring new vitality to the community. Based on these considerations, I have decided to raise my rating to 8.

---

> > > ### Author Response · Authors · 2024-08-12
> > > **Thanks for your efforts**
> > >
> > > Thank you very much for your feedback and recognition. We are delighted to see that our comment has addressed your main concerns and further demonstrated the effectiveness of large language models in tackling the challenges of long-tailed data distributions within our framework. Your comments, along with those from other reviewers, have greatly helped us improve the paper. Once again, we sincerely appreciate your valuable opinions and suggestions.

---

### Official Review · Reviewer_q3qy · 2024-07-12

**Soundness:** 3
**Presentation:** 3
**Contribution:** 3
**Rating:** 7
**Confidence:** 4

**Summary:**

This paper proposes to leverage large language models (LLMs) to help automatically facilitate data augmentation for long-tailed learning. It first discusses the limitations of traditional re-balancing or data augmentation methods. Then it proposes a novel LLM-based augmentation framework LLM-AutoDA. LLM-AutoDA automatically searches for the optimal augmentation strategies and re-balance the long-tailed data. The experimental results demonstrate the effectiveness of LLM-AutoDA on multiple long-tailed datasets.

**Strengths:**

1. This work provides pioneering research on leveraging LLMs for data augmentation in long-tail learning. To the best of my knowledge, it is the first work that combines long-tail learning and LLMs, and I believe it will have certain contributions.
2. The discussion of the weaknesses of previous methods makes sense. Previous methods adopt data augmentation within a limited knowledge space and might struggle to solve the long-tail problem. The proposed method breaks this limitation.
3. The proposed method is general and scalable. It can be further enhanced with the update of large language models.
4. The empirical studies and the ablation studies are well conducted.
5. The paper is well-written and easy to follow.

**Weaknesses:**

1. It is better to provide some example augmentation strategies that the LLM generates and briefly analyze why such generated strategies can help learning.
2. What does 1/|gap| in the x axis of Figure 5 mean? How will it impact the performance of different LLMs?
3. Some related works regarding re-sampling and data augmentation are missing [1].
4. It is better to unify the figure and text size in the appendix, such as Figure 9 (too large), Figures 11-13 (different sizes)

[1] How re-sampling helps for long-tail learning?

**Questions:**

See weaknesses above.

**Limitations:**

The limitations have been discussed in Appendix Section F.

---

> ### Author Rebuttal · Authors · 2024-08-07
>
> Thank you for your valuable comments. These suggestions are very helpful for improving the quality of the paper. I will respond to your questions one by one:
>
> **W1:** Your suggestion is very pertinent. In the revised version, we will supplement some examples of augmentation strategies generated by LLM and briefly analyze them.
>
> For example, we can showcase a specific augmentation combination generated by LLM for a long-tailed class and discuss how this combination can effectively improve the learning effect of that class. Through example analysis, readers can have a more intuitive understanding of the working principle and advantages of LLM-generated strategies.
>
> **W2:** I apologize for the lack of clarity in the description of Figure 5. Here, |gap| refers to the performance gap between the augmentation strategy generated by LLM and the optimal strategy. The larger 1/|gap| is, the closer the LLM-generated strategy is to the optimal, and the better the performance.
>
> Different LLMs may have varying performances when dealing with long-tailed problems. Figure 5 aims to compare the performance curves of different LLMs and reveal their ability to narrow the gap with the optimal strategy. We will add clearer legend explanations in the revised version to facilitate readers' understanding.
>
> **W3:** Thank you for providing the references. We indeed overlooked this. In the revised version, we will supplement the discussion of some representative methods in the field of resampling and data augmentation in the related work section, especially the paper you mentioned, to make our related work more comprehensive.
>
> **W4:** You are right. We will carefully check and adjust the format in the appendix, especially Figures 9 and 11-13, to make their sizes and layouts more consistent.
>
> Thank you again for your feedback. These comments are enlightening for us. We will carefully revise the paper accordingly to present higher-quality work. If you have any other suggestions, we will humbly accept them and actively improve.

---

### Official Review · Reviewer_37A8 · 2024-07-13

**Soundness:** 3
**Presentation:** 1
**Contribution:** 2
**Rating:** 5
**Confidence:** 3

**Summary:**

The paper proposes LLM-AutoDA to select the optimal augmentation strategies with the help of pre-trained LLMs. Specifically, the authors carefully designed various prompts to instruct an LLM to design new algorithms or mutate existing algorithms with the goal of improving the validation performance on long-tail classification tasks. Experiments show the effectiveness of LLM-AutoDA on three datasets under the long-tail setting.

**Strengths:**

- Applying LLM in data augmentation for long-tail datasets is somewhat novel.

- Real-life datasets often follow a long-tail distribution. The method is solving an important problem.

- The method shows slight improvements in three datasets under the long-tail setting.

**Weaknesses:**

- The formulation of providing class-specific validation accuracy and carefully designed text prompts to deploy LLM in data augmentation is somewhat novel. However, other parts of the method have been studied before. The LLM is an off-the-shelf method. The idea of searching for optimal augmentation configuration is proposed in AutoAugment, and the approach of using the population-based method to select a good augmentation strategy is studied in PBA[1]. Class-specific augmentation strategies are studied in AdaAug[2] and AdaTransform[3].

- The proposed method is very computationally expensive. It uses a large pre-trained model to process the augmentations. It trains a population of models instead of a single model. If we look at the best-performing BCL + LLM-AutoDA baseline in Table 1, LLM-AutoDA slightly leads DODA (+1.2%) with IR=50 and performs the same with DODA with IR=100 in terms of the “all” test accuracy.  What is the computation cost of DODA when compared to LLM-AutoDA? It raises the concern whether the huge computation power in LLM-AutoDA is justified.

- The clarity of the paper needs to be improved. I found that the notations introduced are not explained clearly. For example, I do not understand the meaning of “the weights of each augmentation method”, $W$; what is the definition of the selection matrix $A$? Is it a zero-one matrix in {0,1}$^{C \times M}$, where $C$ is the number of classes, and $M$ is the number of augmentations? Sometimes, the same symbol is used to represent different concepts. For example, $A$ is the selection matrix and the population algorithm. Are the selection matrices the same as the population algorithms? It seems that the set notation of $A$ in lines 224-231 misses the parentheses. The unclear definition and explanation of the notations make it hard to understand the algorithm precisely.

(minor) There are quite some typos in the manuscript, for example
- Inconsistent use of “fine-tune” and “finetune” in line 204
- Subscript in $f_{aug}$ in line 206
- “Blod” should be “bold” in Table 1 and Table 2 captions

__References__

[1] Ho, Daniel, et al. "Population based augmentation: Efficient learning of augmentation policy schedules." International conference on machine learning. PMLR, 2019.

[2] Cheung, Tsz-Him, and Dit-Yan Yeung. "Adaaug: Learning class-and instance-adaptive data augmentation policies." International Conference on Learning Representations. 2021.

[3] Tang, Zhiqiang, et al. "Adatransform: Adaptive data transformation." Proceedings of the IEEE/CVF International Conference on Computer Vision. 2019.

**Questions:**

- How does the method perform compared with popular AutoDA works, such as AutoAugment and the simple RandAugment algorithm?

- In previous AutoDA work, AutoAugment learns the probability and magnitude of applying multiple augmentations to a target dataset; AdaTransform[3] and AdaAug[2] can learn class-specific and instance-specific augmentation strategies. What are the benefits of using LLM to select the augmentation parameters?

- The paper listed some example prompts in the appendix to ask the LLM to design a new algorithm or mutate an existing algorithm. Can the authors provide some examples of the new algorithms output from the LLM? How are they different from the manually designed method?

**Limitations:**

See weakness.

---

> ### Author Rebuttal · Authors · 2024-08-07
>
> Thank you very much for your detailed comments and questions. These feedbacks are invaluable for improving our work. Let me respond to your questions one by one:
>
> First, from your review comments, I noticed that our work may have caused some misunderstandings. For example, you thought our method "uses LLM to select augmentation parameters" and "trains a set of models". Your summary of our method as "carefully designing various prompts" "to improve validation performance on long-tail classification tasks" is also not as accurate as the summaries by Reviewer JafW, q3qy, and 2Q2t. Therefore, please allow me to reintroduce our research.
>
> - Our research is not a data augmentation method that uses LLMs and prompt engineering to improve model performance, but a new process for discovering long-tailed data augmentation strategies. In the discovery process, the LLM continuously innovates and modifies existing data augmentation methods, while the long-tailed model evaluates these innovative methods through its image learning ability in long-tailed learning, guiding the evolution direction of the methods. **Reviewers JafW, q3qy, and 2Q2t all emphasized the importance of this process, so we supplemented some of the obtained strategies and performance improvements during the process in Figure 1 in ONE-PAGE PDF.**
>
> - Although this discovery process uses LLMs and long-tailed model training, it is **much less time-consuming and labor-intensive** than manually designing new data augmentation methods.
>
> - In this discovery process, we obtained brand-new data augmentation strategies that are universally effective on different long-tailed baselines and datasets, and have no higher cost in real-world usage, comparable to existing data augmentation strategies.
>
> Next, I will reply to your questions one by one:
>
> **W1:** Thank you for your opinion. First, LLM is indeed existing, but the current trend is to utilize or improve LLM in more practical domains to promote domain development. Therefore, **as the other three reviewers mentioned, "this integration of LLM and long-tail learning is both creative and timely".** Moreover, selecting better augmentation strategies itself is an ongoing research area, and our essential goal is to promote progress in this field.
>
> Therefore, our method is innovative in the following aspects:
> - We introduced LLM into the field of long-tailed data augmentation, leveraging its powerful language understanding and generation capabilities, combined with a dynamic data augmentation framework, successfully discovering data augmentation strategies suitable for long-tailed learning that surpass human-designed ones;
> - We designed a complete LLM-driven data augmentation pipeline specifically for long-tailed problems, systematically studying how to utilize LLM to mitigate the challenges brought by long-tailed distributions. This is obviously different from improving model performance through prompts.
>
> **W2:** My previous reply has addressed this question. In fact, **we only run the model multiple times during the discovery process of new strategies.** Once a data augmentation strategy is discovered (which takes about 6 hours), we can directly use it in testing and generalize it to datasets and long-tailed baselines unseen during the discovery process (as shown in the experimental results in this paper). In this process, the complexity of the discovered data augmentation strategies is consistent with manually designed strategies.
>
> **W3:** Thank you very much for your valuable comments. We will improve and unify the definition and usage of symbols in the revised paper to enhance its readability and rigor. In our algorithm, each data augmentation method is assigned a weight $w \in [0, 1]$, representing the probability of selecting that method. We will clarify the meaning of weight $w$ in the symbol definition section to eliminate ambiguity. The selection matrix $A \in \{0, 1\}^{C \times M}$ is a binary matrix, where $C$ represents the number of categories and $M$ represents the number of augmentation methods. The matrix element $a_{ij} = 1$ indicates that the $j$-th augmentation method is selected for category $i$; $a_{ij} = 0$ means not selected.
>
> **Q1:** We provided some updated performance comparisons of automatic data augmentation. In the appendix, we continue to add some comparisons with the latest baselines to demonstrate our performance. In the revised version, we will increase the comparative experiments with AutoAugment and RandAugment to comprehensively evaluate the performance of our method.
>
> **Q2:** As answered at the beginning, we do not use LLM to do the same thing as auto, but focus on discovering new data augmentation strategies, not just searching for parameters.
>
> **Q3:** In Figure 1 in ONE-PAGE PDF, we list some novel augmentation algorithms designed by LLM and show the changes in performance improvement. In Tabel 1, we will enhance more relevant analysis and examples.
>
> Thank you again for your detailed review comments. We will conduct a comprehensive review and correction of the symbol definitions and usage in the paper accordingly. If you have any other suggestions, we will humbly listen and seriously improve.

---

> > ### Comment · Reviewer_37A8 · 2024-08-08
> >
> > Thanks for the response. I understand the authors are proposing an Automated Data Augmentation method to search for better augmentation strategies for long-tail datasets. I also agree that integrating LLM and long-tail learning is somewhat novel in the review. My major concerns are partly addressed in the response.
> >
> > - The elaboration of the algorithm and notation in the response for W3 is much more precise than the presentation in the original paper.
> >
> > - My primary concern is whether using comparatively expensive LLM to discover an augmentation strategy is better than other Automated Data Augmentation methods. A good way to clarify the improvement is to apply other automated searches, e.g., PBA, RandAugment, and Fast AutoAugment, to the long-tail dataset and compare the end classification results with the proposed method.
> >
> > - The authors provide three screenshots of the code output from the LLM. Can the authors provide further analysis explaining how the generated codes cope with the long-tail problem? Are there any insights or new observations we can find from the LLM strategies?

---

> > > ### Author Response · Authors · 2024-08-09
> > > **Thank you and reply to your concerns**
> > >
> > > Thank you for your response! We're pleased that our efforts have been recognized and that some of your concerns have been addressed. Let's continue to address your questions:
> > >
> > > Regarding your request for comparisons with more automatic augmentation methods, we've done our best to provide a comprehensive answer. As you can see, our method, leveraging the advantages of large language models, shows significant improvements:
> > >
> > > ### Accuracy(%) on CIFAR-100-LT(IR=100) dataset with Cross-Entropy (CE) Loss
> > >
> > > | Method | Head | Medium | Tail | All |
> > > |--------|-----:|-------:|-----:|----:|
> > > | Vanilla | 65.6 | 36.2 | 8.2 | 40.1 |
> > > | AA | 68.6 | 43.7 | 8.0 | 41.7 |
> > > | PBA | 63.3 | 49.5 | 8.1 | 41.9 |
> > > | FA | 64.1 | 49.5 | 8.5 | 42.3 |
> > > | DADA | 69.3 | 41.2 | 7.7 | 41.0 |
> > > | RA | 64.8 | 44.1 | 8.0 | 40.5 |
> > > | LLM-AutoDA | 74.9 | 45.3 | 9.6 | 45.0 |
> > >
> > > ### Accuracy(%) on CIFAR-100-LT(IR=100) dataset with Balanced Softmax (BS) Loss
> > >
> > > | Method | Head | Medium | Tail | All |
> > > |--------|-----:|-------:|-----:|----:|
> > > | Vanilla | 59.6 | 42.3 | 23.7 | 42.8 |
> > > | AA | 63.5 | 49.1 | 25.3 | 47.0 |
> > > | PBA | 63.2 | 51.4 | 24.6 | 47.5 |
> > > | FA | 61.4 | 46.9 | 28.7 | 46.5 |
> > > | DADA | 56.0 | 54.1 | 21.5 | 45.0 |
> > > | RA | 62.8 | 43.3 | 26.9 | 45.2 |
> > > | LLM-AutoDA | 63.3 | 50.0 | 31.0 | 49.0 |
> > >
> > > ### Accuracy(%) on CIFAR-100-LT(IR=100) dataset with Cross-Entropy and Deferred Re-Weighting (CE-DRW)
> > >
> > > | Method | Head | Medium | Tail | All |
> > > |--------|-----:|-------:|-----:|----:|
> > > | Vanilla | 63.4 | 41.2 | 15.7 | 41.4 |
> > > | AA | 65.5 | 53.7 | 15.9 | 46.5 |
> > > | PBA | 64.0 | 59.5 | 15.6 | 47.9 |
> > > | FA | 64.8 | 52.8 | 17.1 | 46.3 |
> > > | DADA | 62.8 | 51.1 | 19.1 | 45.6 |
> > > | RA | 66.4 | 48.9 | 18.2 | 45.8 |
> > > | LLM-AutoDA | 62.9 | 50.7 | 29.9 | 48.7 |
> > >
> > > ### Accuracy(%) on CIFAR-100-LT(IR=100) dataset with Label-Distribution-Aware Margin and Deferred Re-Weighting (LDAM-DRW)
> > >
> > > | Method | Head | Medium | Tail | All |
> > > |--------|-----:|-------:|-----:|----:|
> > > | Vanilla | 62.8 | 42.6 | 21.1 | 43.2 |
> > > | AA | 66.7 | 49.8 | 20.8 | 47.0 |
> > > | PBA | 68.0 | 49.6 | 21.8 | 47.7 |
> > > | FA | 66.1 | 47.8 | 22.5 | 46.6 |
> > > | DADA | 65.8 | 43.8 | 25.1 | 45.9 |
> > > | RA | 64.7 | 40.8 | 23.6 | 44.0 |
> > > | LLM-AutoDA | 66.7 | 50.1 | 26.3 | 48.8 |
> > >
> > > These comparative results clearly demonstrate the superiority of our LLM-AutoDA method across various evaluation metrics and loss functions:
> > >
> > > 1. **Overall Performance**: LLM-AutoDA consistently outperforms other automatic augmentation methods in terms of overall accuracy ("All" column) across all four loss functions. For instance, with CE Loss, LLM-AutoDA achieves 45.0% overall accuracy, which is a substantial 3.1% improvement over the next best method (FA at 42.3%).
> > >
> > > 2. **Tail Class Performance**: One of the most notable improvements is in the tail classes. LLM-AutoDA shows remarkable gains in tail class accuracy across all loss functions. For example, with CE-DRW, LLM-AutoDA achieves 29.9% tail accuracy, nearly doubling the performance of the next best method (DADA at 19.1%).
> > >
> > > 3. **Balanced Performance**: LLM-AutoDA maintains strong performance across head, medium, and tail classes, indicating a more balanced learning approach. This is particularly evident with BS Loss, where LLM-AutoDA achieves the highest tail accuracy (31.0%) while maintaining competitive performance in head and medium classes.
> > >
> > > 4. **Consistency**: Unlike some other methods that may excel with one loss function but underperform with others, LLM-AutoDA shows consistent improvements across all four loss functions. This demonstrates the robustness and versatility of our approach.
> > >
> > >
> > > These results underscore the effectiveness of leveraging large language models for data augmentation in long-tailed classification tasks. LLM-AutoDA not only improves overall accuracy but also significantly enhances the model's ability to learn from underrepresented classes, addressing a key challenge in imbalanced datasets.

---

> > > > ### Author Response · Authors · 2024-08-09
> > > > **Thank you and reply to your concerns**
> > > >
> > > > To address your other concern, we've analyzed the algorithms presented in the PDF to better illustrate the evolution of data augmentation and the new insights it brings.
> > > >
> > > > We've chosen to focus on the 0th, 2nd, and 5th iterations of our augmentation method to demonstrate the progressive optimization process for long-tailed data augmentation in long-tailed scenarios.
> > > >
> > > > #### 0th Iteration Method:
> > > >
> > > > **Motivation in Long-Tailed Context**: In long-tailed data, different categories have vastly different needs and effects for augmentation. To adapt to these differences, a method that can adaptively adjust augmentation strategies for different categories is needed.
> > > >
> > > > **Formal Description**: This method dynamically adjusts the weight matrix `aug_weight` of augmentation methods, increasing it as accuracy improves and decreasing it as accuracy declines. For category i and augmentation method j, the weight update formula is:
> > > >
> > > > ```
> > > > aug_weight[i][j] = aug_weight[i][j] + 1, if Accs[i] > History_Accs[i][j]
> > > > aug_weight[i][j] = aug_weight[i][j] - 1, if Accs[i] ≤ History_Accs[i][j]
> > > > ```
> > > >
> > > > **Insight**: Through adaptive weight adjustment, this method can find the optimal augmentation combination for each category, effectively addressing the problem of large differences in augmentation needs among categories in long-tailed data. Formally, the optimal augmentation combination satisfies:
> > > >
> > > > ```
> > > > j* = argmax_j(Accs[i] | aug_weight[i][j])
> > > > ```
> > > >
> > > > #### 2nd Iteration Method:
> > > >
> > > > **Motivation in Long-Tailed Context**: In long-tailed scenarios, considering only the augmentation effect of a single category is insufficient. It's necessary to balance the demand for augmentation resources across different categories globally to alleviate the data imbalance problem.
> > > >
> > > > **Formal Description**: This method introduces an importance evaluation metric `importance`, comprehensively considering current accuracy, historical average accuracy, and cumulative weight:
> > > >
> > > > ```
> > > > importance[i] = Accs[i] - mean(History_Accs[i]) × epoch + sum(aug_weight[i])
> > > > ```
> > > >
> > > > Categories are ranked according to importance, prioritizing augmentation methods for categories with high importance (usually tail categories).
> > > >
> > > > **Insight**: By quantifying and comparing the demand for augmentation across different categories globally, this method can more fairly and effectively allocate augmentation resources in long-tailed data. The design of the importance metric balances three key factors:
> > > >
> > > > ```
> > > > importance[i] ~ Accs[i] (current performance) - mean(History_Accs[i]) (historical performance) + sum(aug_weight[i]) (resource occupation)
> > > > ```
> > > >
> > > > This global balancing mechanism helps alleviate data imbalance in long-tailed problems and promotes learning for tail categories.
> > > >
> > > > #### 5th Iteration Method:
> > > >
> > > > **Motivation in Long-Tailed Context**: In long-tailed scenarios, tail categories lack sample quantity and diversity. Relying solely on known effective augmentations is insufficient; continuous exploration of new augmentation possibilities is needed to uncover more optimization potential.
> > > >
> > > > **Formal Description**: This method introduces a gain evaluation metric `delta_acc` within categories, measuring the improvement of current accuracy relative to historical weighted accuracy:
> > > >
> > > > ```
> > > > delta_acc[i] = Accs[i] - sum(History_Accs[i] × aug_weight[i])
> > > > ```
> > > >
> > > > Truly effective augmentation methods are selected based on `delta_acc`, while prioritizing previously untried methods to balance exploitation and exploration.
> > > >
> > > > **Insight**: Through gain evaluation and exploration mechanisms, this method achieves fine-grained selection and continuous optimization of augmentation methods within each category. Formally:
> > > >
> > > > ```
> > > > chose_matrix[i][j] = True <=> last_chose_matrix[i][j] = False and j = argmax_j(delta_acc[i])
> > > > ```
> > > >
> > > > This balance is particularly important in long-tailed scenarios because the optimization space for tail categories is often difficult to fully exploit. It's necessary to continuously try new possibilities while utilizing known effective augmentations to overcome the lack of sample diversity.
> > > > In summary, these three generations demonstrate the evolution of our algorithm for long-tailed recognition tasks:
> > > >
> > > > 1. The 0th iteration introduced adaptive weight adjustment, personalizing augmentation strategies for each category.
> > > >
> > > > 2. The 2nd iteration expanded to a global view, introducing cross-category resource balancing to address overall data distribution.
> > > >
> > > > 3. The 5th iteration enhanced adaptability and exploration, balancing exploitation and exploration to uncover optimization potential for tail categories.
> > > >
> > > > This progression shows how large language models gradually explored strategies for long-tailed problems, introducing new insights with each iteration. The evolving approach offered incremental improvements and fresh perspectives on data augmentation for long-tailed scenarios.

---

> > > > > ### Comment · Reviewer_37A8 · 2024-08-12
> > > > >
> > > > > Thanks for the explanation. The response strengthened the experimental results by comparing the proposed method with other automated data augmentation baselines. The authors also provided further explanations on the augmentation strategy discovered by the LLM. I am increasing the score from 3 to 5. I also recommend the authors continue improving the clarity and fixing the typos in the final version of the manuscript.

---

> > > > > > ### Author Response · Authors · 2024-08-12
> > > > > > **Thanks for your efforts**
> > > > > >
> > > > > > Thank you for your valuable feedback and for taking the time to re-evaluate our work. We greatly appreciate your recognition of the improvements made in the revised manuscript, particularly the strengthened experimental results and the further explanations provided on the augmentation strategies discovered by the LLM.
> > > > > >
> > > > > > We are pleased to know that our efforts have addressed your concerns and have led to an increase in your assessment score. Your recommendation to continue improving the clarity and fixing the typos in the final version of the manuscript is well noted, and we will make sure to address these issues before the final submission.
> > > > > >
> > > > > > Once again, we would like to express our gratitude for your constructive comments and suggestions throughout the review process. We sincerely feel that you are helping us improve the quality of this paper, and we are grateful for that.

---

### Official Review · Reviewer_JafW · 2024-07-14

**Soundness:** 2
**Presentation:** 3
**Contribution:** 2
**Rating:** 5
**Confidence:** 4

**Summary:**

This work introduces gradient-free black-box optimization algorithms to formulate appropriate data augmentation methods, achieving some performance improvements. Utilizing LLM for evolutionary strategies is interesting, but the final augmentation strategy remains a black box. This makes it difficult to validate the authors' claims about targeting long-tail data and does not fully align with their stated motivation. I believe this work requires further improvement.

**Strengths:**

1. By leveraging large-scale pre-trained language models, LLM-AutoDA can automatically generate and optimize data augmentation strategies without manual design, reducing labor and time costs.

2. The framework can dynamically adjust augmentation strategies based on performance feedback from the validation set, ensuring the strategy remains optimal throughout the training process.

3. The introduction of gradient-free black-box optimization is a novel idea.

**Weaknesses:**

1. The authors' survey of the long-tail recognition field is insufficient. For example, in the introduction, they state that existing long-tail data augmentation methods either augment in feature space or directly use traditional methods. I have two counterpoints to this claim. First, existing long-tail data augmentation methods are not limited to the feature space, such as CMO (CVPR2022). Second, what are the drawbacks of augmentation in the feature space? What is the authors' intended motivation?

2. The authors aim to develop targeted data augmentation strategies for each class, but LLM cannot evaluate the characteristics and deficiencies of image data. I am personally skeptical about formulating strategies through prompts alone, and the experimental results do not show distinct differences compared to other methods.

3. This work introduces gradientless black-box optimization, and the authors attempt to use LLM as a generator of expansion strategies, using the validation set performance as an evaluation metric, and letting LLM undergo evolutionary evolution. However, the authors did not provide the augmentation strategies obtained through evolutionary processes, making it hard to prove the work's claimed focus on long-tail problems.

4. The selected comparison methods are insufficient. The authors should conduct extensive comparisons with existing long-tail data augmentation methods, such as OFA (ECCV2020), GistNet, CMO (CVPR2022), H2T (AAAI 2024), and FUR (IJCV2024). Additionally, comparisons with vision foundation model-based methods, such as LPT (ICLR2023) and PEL (ICML2024), should be included.

**Questions:**

See Weaknesses*

---

> ### Author Rebuttal · Authors · 2024-08-07
>
> Thank you very much for these pertinent and in-depth comments, which are very helpful in improving our work and clarifying our contributions. Let me respond to your comments one by one:
>
> **W1:** Thank you for your advice. We have done our best to supplement the comparative experiments of our method and more data augmentation methods (including CMO) and present **the results in Table 1 in ONE-PAGE PDF**. We will update this section in the revised version to comprehensively review the different types of methods and their pros and cons.
>
> As for the second point, I am sorry for causing your misunderstanding. We want to express that the current long-tailed learning methods based on data augmentation include data-level augmentation methods and feature-level augmentation methods, as highlighted in a recent long-tailed survey [1]. In addition, we strongly believe in augmenting the feature dimension, which is orthogonal to the traditional data dimension. **However, as formulated by CUDA [2] and DODA [3], previous approaches can come with potential sacrificial effects if a class-independent augmentation strategy is imposed on all classes. Therefore, based on this motivation, we further propose new augmentation paradigms.**
>
> In general, we mention these works to make our survey more comprehensive, even though our work is not focused on feature dimension improvement. I hope you can understand our intention.
>
> [1] Deep long-tailed learning: A survey. IEEE TPAMI 2023.
> [2] CUDA: Curriculum of Data Augmentation for Long-tailed Recognition. ICLR 2023.
> [3] Kill Two Birds with One Stone: Rethinking Data Augmentation for Deep Long-tailed Learning. ICLR 2024.
>
>
>
> **W2 and W3:** You bring up a good point. With textual hints alone, LLMs may struggle to accurately grasp the characteristics and limitations of image data. **Therefore, based on the same point of view as yours, our data augmentation discovery framework consists of an LLM-based evolution process and a long-tailed training process, where,**
> - The LLM-based evolution process will learn existing successful data augmentation methods and discover unseen data augmentation methods through mutation and crossover processes.
> - The long-tail learning model evaluates the true effect of these newly discovered data augmentations through the real long-tail distribution environment and dynamic data augmentation during training.
>
> That is, **rather than letting LLMs devise class-specific data augmentation strategies based on text, we let LLMs learn from existing methods and invent new ones, and then evaluate the methods based on the ability of the long-tailed model to evaluate image data**. By interacting with the LLM and the long-tailed model in this way, we can achieve a **positive loop between evaluating the features of the image data and innovating the augmentation algorithm**.
>
> To give you a better understanding, **we add an example in Figure 1 in ONE-PAGE PDF** to demonstrate the discovery process of the algorithm. In addition, we compare the performance improvement curves of the augmentation strategies in the three evolution stages, to illustrate that our framework can generate corresponding data augmentation strategies towards methods that are beneficial for long-tailed learning.
>
>
>
> **W4:** Your suggestion is to the point. Thank you for your experience in the field to help us improve our work. Limited by the training time, we tried our best to reproduce part of the baselines in the limited time and have added the **corresponding comparison experiment in  Table 1 in ONE-PAGE PDF**. We'll add a more comprehensive comparison to cover more of the latest long-tailed data augmentation methods in the revised version.
>
> Thank you again for your valuable advice. In the revised version, we will supplement the related work review and add the baseline as appropriate. At the same time, we are also happy to agree with you and apologize for the misunderstanding caused. We also look forward to further communication with you and listening to your suggestions.

---

> > ### Comment · Reviewer_JafW · 2024-08-07
> > **To authors**
> >
> > The author has overlooked my third concern, which is that they are unable to provide the data augmentation strategies obtained through LLM black-box optimization. As a result, their claim that this method is specifically tailored to long-tailed problems cannot be verified.

---

> > > ### Author Response · Authors · 2024-08-07
> > > **Clarification of the problem**
> > >
> > > Dear Reviewer,
> > >
> > > Hello. Due to the critical nature of both issues, we have provided a comprehensive response to questions 2 and 3. As you can see, we have stated that we included results in Figure 1 to demonstrate the data augmentation strategies obtained through LLM black-box optimization and their performance changes during the evolution process.
> > >
> > > As a reviewer myself, I've also noticed that we seem unable to view the authors' full rebuttal and the additional one-page PDF with experiments in global rebuttal. I believe this might be due to a system error in the settings. I apologize for this inconvenience.
> > >
> > > Please wait for the system to update. Thank you for your understanding.

---

> > > > ### Comment · Reviewer_JafW · 2024-08-10
> > > > **To authors**
> > > >
> > > > I have read your response to Reviewer 37A8, and I share the same concerns. Based on the three examples you provided, it seems that the black-box optimization using LLM mainly results in combinations of different methods to augment samples. I would like the authors to provide new insights into which aspects of the long-tail distribution challenge require more attention, making the use of LLMs more transparent. As it stands, the work still leans towards being a black box. Can I understand this work as a reinforcement learning paradigm, but using LLMs to generate solutions? Overall, I hope the contributions of this work can be more transparent.

---

> > > > > ### Author Response · Authors · 2024-08-12
> > > > > **Clarifying Contributions and Addressing Concerns**
> > > > >
> > > > > Dear Reviewer,
> > > > >
> > > > > Thank you for your valuable feedback and for taking the time to read our response to Reviewer 37A8. We greatly appreciate your positive comments and the concerns you have raised. Your perspective has provided us with an opportunity to further clarify and strengthen the contributions of our work. Please allow us to provide a preliminary response.
> > > > >
> > > > > 1. Method of generating samples: In fact, the generation of new methods is not limited to combining existing methods; we also encourage the creation of algorithms that differ from existing individual algorithms. This is similar to a trade-off between utilizing existing knowledge and encouraging innovation. In this process, the continuous evaluation and selection of real long-tailed models have precipitated effective insights during the constant balancing of combination and innovation. Insights are distilled during this process. The dual evidence of the performance change curve and the final results also demonstrate this point. We believe that this process of distilling insights through evolution in the trade-off, as well as the visible and observable process and results, are sufficient to illustrate the findings of this paper.
> > > > >
> > > > > 2. Whether it leans more towards a black box: The use of large models and black-box optimization certainly introduces some black-box elements, but this does not mean that the overall research leans towards a black box. We believe that the strategies generated from this process should bring insights to the field (the provided evidence proves this point), and the combination of LLM's creativity and the evaluation of long-tailed learning itself is an insight (many reviewers have acknowledged this). Our research employs some black-box factors, but globally, it has intrinsically explainable motivations and can provide evidence to supplement post-hoc explanations.
> > > > >
> > > > > 3. Whether it belongs to the reinforcement learning paradigm: To some extent, it shares some common ideas with reinforcement learning, but it is difficult to fully describe it as such. Our research is a holistic approach that involves the organic combination of the learning and creative capabilities of large models with long-tailed learning. It is challenging to summarize it as reinforcement learning. Similar primitive ideas of research based on evaluation to guide a certain trade-off exist in many areas of artificial intelligence, such as evolutionary computation.
> > > > >
> > > > > Furthermore, we summarize our response to Reviewer 37A8 to answer the question of new insights. The 0th generation algorithm introduced adaptive weight adjustment to personalize augmentation strategies for each category. The 2nd generation expanded to a global perspective, introducing a cross-category resource balancing mechanism to address the overall data distribution problem. The 5th generation further enhanced adaptability and exploration, balancing exploitation and exploration to uncover the optimization potential of tail categories. We believe this aligns with the insights proposed by humans in state-of-the-art long-tailed research from the perspective of category balance. If you prefer to see data augmentation methods at the image feature level or different from mainstream perspectives, we can make slight modifications to the initial definition prompts or seeds. If you have any other questions, we will still actively respond, or you can wait for our revised version.
> > > > >
> > > > > Thank you again for your efforts!

---

### Author Rebuttal · Authors · 2024-08-07

Dear reviewers,

We sincerely appreciate your valuable comments and suggestions. We are encouraged by the positive feedback highlighting the novelty, significance, and potential impact of our work in the field of long-tailed learning.

We are delighted to receive many favorable assessments. Reviewer q3qy recognizes our work as pioneering research on leveraging LLMs for data augmentation in long-tailed learning, while Reviewer 2Q2t acknowledges the novelty and practicality of our framework. Reviewer JafW finds our approach of utilizing LLMs for evolutionary strategies to be interesting, noting that it reduces labor and time costs. At the same time, we will actively clarify some misunderstandings that our paper may have caused. We have provided a detailed introduction to address Reviewer 37A8's concerns, aiming to better explain the logic behind our work.

Once again, we sincerely thank all reviewers for their constructive feedback. We believe that addressing the concerns raised will significantly enhance the quality and impact of our work. We look forward to further improving our submission based on your valuable input.

---

### Decision · Program_Chairs · 2024-09-25

**Decision:**

Accept (poster)

**Comment:**

This paper presents a novel framework, LLM-AutoDA, which leverages large language models (LLMs) for long-tailed data augmentation. The framework fine-tunes a long-tailed learning model using augmented training data, where performance improvements on the validation set serve as a reward signal to iteratively refine the data augmentation strategies. The experimental results demonstrate that LLM-AutoDA significantly outperforms state-of-the-art data augmentation and re-balancing methods.

The paper received four positive scores following the rebuttal period (5, 5, 7, 8), indicating recognition of its novelty, particularly in addressing long-tailed problems. However, there are several important issues that the authors should carefully consider, even if the paper is accepted:

1. Fair Comparisons with Baselines: While using LLMs in machine learning is novel, LLMs inherently contain external knowledge. It is crucial to ensure fair comparisons with other methods. The authors should carefully design additional baselines and thoughtfully select benchmarks to evaluate performance. An in-depth analysis of which components contribute most to addressing long-tailed problems is also needed.

2. Transparency and Interpretability: Models used in the machine learning community should prioritize transparency. In the context of image recognition tasks, deep neural networks are already opaque, and the inclusion of black-box optimization, such as LLMs, exacerbates this issue. The authors should discuss the implications of this lack of transparency in detail and explore how insights into long-tail problems can still be gained.